# YTHDC1 delays cellular senescence and pulmonary fibrosis by activating ATR in an m6A-independent manner

Canfeng Zhang [1,2,5], Liping Chen [3,5], Chen Xie[1], Fengwei Wang[1], Juan Wang[4], Haoxian Zhou [1], Qianyi Liu [1], Zhuo Zeng[1], Na Li[1], Junjiu Huang[1], Yong Zhao [1,6] & Haiying Liu [1✉]

## Abstract

**Accumulation of DNA damage in the lung induces cellular senescence and promotes age-related diseases such as idiopathic pulmonary fibrosis (IPF). Hence, understanding the mechanistic regulation of DNA damage repair is important for anti-aging therapies and disease control. Here, we identified an m6A-independent role of the RNA-binding protein YTHDC1 in counteracting stress-induced pulmonary senescence and fibrosis. YTHDC1 is primarily expressed in pulmonary alveolar epithelial type 2 (AECII) cells and its AECII expression is significantly decreased in AECIIs during fibrosis. Exogenous overexpression of YTHDC1 alleviates pulmonary senescence and fibrosis independent of its m6A-binding ability, while YTHDC1 deletion enhances disease progression in mice. Mechanistically, YTHDC1 promotes the interaction between TopBP1 and MRE11, thereby activating ATR and facilitating DNA damage repair. These findings reveal a non-canonical function of YTHDC1 in delaying cellular senescence, and suggest that enhancing YTHDC1 expression in the lung could be an effective treatment strategy for pulmonary fibrosis.**

**Keywords** YTHDC1; ATR; Senescence; IPF
**Subject Categories** DNA Replication, Recombination & Repair; Respiratory System

## Introduction

Idiopathic pulmonary fibrosis (IPF) is a progressive, irreversible and eventually fatal interstitial lung disease (Richeldi et al, 2017). Currently, there is still a lack of an effective medical therapy for IPF because of its unclear underlying cellular and molecular mechanisms (Wuyts et al, 2013; Wynn, 2011). Considerable evidences suggest that accumulated DNA damage and compromised DNA repair activity,

which induce senescence and destroy function of lung cell, such as alveolar epithelial type II cells (AECII), fibroblasts, and myofibroblasts, appear to be one of the risk factors and major drivers of this disease (Chuang et al, 2013; Hong et al, 2022; Wuyts et al, 2013). Firstly, as compared with healthy people, the lung tissues from IPF patients have more γH2AX signals and telomere uncapping (Schuliga et al, 2021; Wang et al, 2020). Secondly, loss of DNA damage repair factors, such as DNA-PKcs (Habiel et al, 2019), RAD51 (Im et al, 2018), cGAS (Schuliga et al, 2021), and SIRT1 (Zhang et al, 2021), have been implicated in the development of IPF. Moreover, loss or impair the activation of the pivotal DNA damage response (DDR) element, ataxia telangiectasia-mutated and RAD3-related (ATR), induces genome instability, accelerates pulmonary fibrosis (Kumar et al, 2017). In addition, clearance of senescent cells (p16-positive) improved phenotypes in bleomycin-induced lung fibrosis (Schafer et al, 2017). Therefore, it is urgent to explore the exact molecular mechanism of DNA damage repair in lung cells during IPF progression, and pave the way for the development of new therapeutic treatments.

YTHDC1, containing two intrinsically disordered regions (N-terminal and C-terminal domain) and a well-structured m6A binding motif (the YTH domain), recognizes and binds to the m6A modified RNA to regulate RNA metabolism and therefore participates in a variety of cell physiological and pathological processes (Wang et al, 2020; Widagdo et al, 2022; Xu et al, 2014; Yan et al, 2022). On the basis of the conservation of the m6A-interacting residues, the prevailing model has been that YTHDC1 exerts its effects by recognizing m6A through YTH domain (Lee et al, 2021). Our previous study has found that YTHDC1 binds to and stabilizes m6A RNA at DNA double-stranded break (DSB) sites and promotes DNA damage repair (Zhang et al, 2020). Another study found that YTHDC1 binds to m6A-modified EGF transcript and promotes its expression which further upregulated RAD51 (Wang et al, 2022). It is also reported that YTHDC1 promotes cell proliferation in acute myeloid leukemia through binding to m6A-modified MCM4 mRNA and regulating DNA replication (Sheng et al, 2021). These observations imply YTHDC1 involves in various biological processes through recognizing m6A-modified RNAs and regulating their metabolism.

[1]MOE Key Laboratory of Gene Function and Regulation, State Key Laboratory of Biocontrol, School of Life Sciences, Sun Yat-sen University, Guangzhou 510006, China. [2]Center for Translational Medicine, The First Affiliated Hospital, Sun Yat-Sen University, Guangzhou 510080, China. [3]The Center for Medical Research, The First People's Hospital of Nanning City, Nanning 530021, China. [4]Division of Pulmonary and Critical Care Medicine, The First Affiliated Hospital, Sun Yat-sen University, Guangzhou 510080, China. [5]These authors contributed equally: Canfeng Zhang, Liping Chen. [6]Deceased: Yong Zhao. ✉E-mail: liuhy5@mail.sysu.edu.cn

Interestingly, in this study, we uncovered a new function of YTHDC1 independent of its canonical role as an m6A reader. We observed decreased YTHDC1 expression in murine and human fibrotic lungs, which leads to ATR inactivation and DNA damage accumulation. Ectopic expression of YTHDC1 alleviates the pulmonary senescence and fibrosis, whereas knockdown of YTHDC1 aggravates this process. Mechanistically, YTHDC1 promotes the recruitment of TopBP1 to MRE11 through its N terminal domain and activates ATR. Our findings here revealed a noncanonical role for YTHDC1 in preventing the development of pulmonary senescence and fibrosis through activating ATR to timely repair DNA damage and maintain genomic stability.

## Results

### YTHDC1 protects AECII cells from stress-induced senescence and delays lung fibrosis in mice

To explore the role of YTHDC1 in IPF, we constructed pulmonary fibrosis mice model by intratracheal instillation of bleomycin (BLM), which is the most extensively used animal model that recapitulates critical features of human IPF (Liu et al, 2017), and detected the protein levels of YTHDC1 in lung tissue by immunofluorescence (IF). Strikingly, YTHDC1 primarily expresses in AECII (SPC positive) cells in normal lung tissues and significantly decreases during lung fibrosis (Fig. 1A,B). To verify the special change of YTHDC1 expression level in AECII cells, we isolated the AECII cells and lung fibroblasts from BLM-treated and control mice lung to measure the level of YTHDC1. The results show that the level of YTHDC1 specially decreased in AECII but not in fibroblasts (Appendix Fig. S1A–D). Then, we analyzed the YTHDC1 expression level in human IPF patients using RNAseq data downloaded from Gene Expression Omnibus (GEO) database. Data ref (Ahangari et al, 2019). The results show that YTHDC1 decreased significantly in IPF patients (Fig. 1C). We also analyzed the mRNA expression levels of other members of m6A writers, erasers, and readers in IPF patients and found their expression levels almost did not change, with only WTAP and FTO downregulated slightly (Fig. EV1A,B). However, the protein levels of WTAP and FTO were not changed totally or in AECII cells in bleomycin-induced mouse model as compared with control group (Fig. EV1C,D). These results suggest that YTHDC1 but no other members of the RNA m6A pathway decreased during lung fibrosis.

Several Studies have shown that the senescence of AECs (primarily AECII), fibroblasts, and myofibroblasts are involved in the occurrence and development of pulmonary fibrosis. It is reported that senescence of AECII, as the main source of pro-fibrogenic cytokines, serves as an early initiating event in pulmonary fibrosis that leads to fibro-proliferation and progressive loss of lung function (Selman et al, 2001; Selman and Pardo, 2020). In addition, according to our observation above, the expression level of YTHDC1 significantly decreased in AECII of bleomycin treated mice lung. We focused on the role of YTHDC1 in stress-induced AECII senescence and bleomycin-induced mouse pulmonary fibrosis. Firstly, we studied the effect of YTHDC1 depletion on stress-induced cellular senescence. We knocked down YTHDC1 in a rat AECII cell line L2 (Hoffmann et al, 1995; Jiang et al, 2017;

Liu et al, 2016), and observed exacerbated senescence following BLM treatment as evidenced by increased senescence-associated β-galactosidase (SA-β-gal) positive cells (Fig. 1D,E) and decreased cell proliferation (assessed by Ki67) (Fig. EV1E,F). The mRNA levels of p21, p16 and senescence-associated secretory phenotype (SASP) mediators (IL-6, IL-1α, IL1-β, and TNF-α) were also increased when compared to the NC group (Fig. EV1G). In addition, we treated the YTHDC1 depleted L2 cells with VP-16, another drug causes DNA damage and induces senescence (Zhang et al, 2021), and also got the same results (Fig. EV1H,I). Then, we studied the effect of YTHDC1 depletion on senescence and lung fibrosis in mice. We knocked down YTHDC1 in mice lungs using Adeno-associated virus serotype 6 (AAV6) expressing system and then subjected the mice to BLM or saline injection 7 days later (Fig. 1F). Lung tissues were harvested for experiments 7 days after BLM or saline delivery, as the aging level of AECII is the highest at this time point (Aoshiba et al, 2003). It reveals that depletion of YTHDC1 significantly deteriorates the BLM induced decline of lung physiological function, pulmonary senescence, fibrosis and DNA damage in mice lungs, as evidenced by the increase of lung density (Fig. 1G,H), SASP (Fig. 1I), SA-β-gal in AECII (co-staining with SPC (brown signal)) (Fig. 1J,K), the senescence marker p21 and p16 (Fig. EV1J–M), Masson's trichrome staining (Fig. 1L,M), another marker of fibrosis α-SMA (Fig. EV1N,O), and DNA damage marker γH2AX (Fig. 1N,O). In addition, not only the γH2AX positive cell but also the foci number of γH2AX increased after depletion of YTHDC1 (Fig. EV1P–R). As 14 days after bleomycin instillation is the best time point to measure lung fibrosis parameters, we performed the masson's staining and α-SMA IHC to detect the lung fibrosis in mice lung transfected with shYTHDC1 or control 14 days after BLM treatment and also got the same conclusion (Appendix Fig. S2A–D). Altogether, these results display that depletion of YTHDC1 promotes stress-induced senescence and lung fibrosis, supporting the idea that YTHDC1 plays a protective role during pulmonary senescence and fibrosis progress.

To further verify the role of YTHDC1 in the process of pulmonary senescence and fibrosis, experiments with YTHDC1 overexpression have been performed. We overexpressed wild type (WT) or mutant (MUT, without m6A binding activity) YTHDC1 in L2 cells and treated the cells with BLM. Unexpectedly, the result shows that overexpression of either YTHDC1-WT or YTHDC1-MUT decreases the levels of SA-β-gal, p21, and p16 (Figs. 2A,B and EV2A) and increases the Ki67-positive cells (Fig. EV2B,C). Furthermore, simultaneous knockdown of YTHDC1 and METTL3 further aggravated the senescence induced by YTHDC1 or METTL3 deletion alone (Figs. 2C,D and EV2D). These results suggest that YTHDC1 protects cells from stress-induced senescence independent of m6A pathway, and the effect of METTL3 deletion is consistent with previous report that METTL3-m6A-IGF2BP2 axis counteracts premature aging via stabilization of MIS12 mRNA (Wu et al, 2020). We also overexpressed wild type or mutant YTHDC1 in pulmonary fibrosis mice model using AAV system (Fig. 2E). Consistent with above observations, the senescence and fibrosis induced by BLM were inhibited by overexpression of YTHDC1, both wild type and mutated ones (Figs. 2F–M and EV2E–G, and Appendix Fig. S2E–H). Remarkably, the co-staining of p16 or p21 with SPC and YTHDC1 show that overexpression of YTHDC1-WT or –MUT specifically

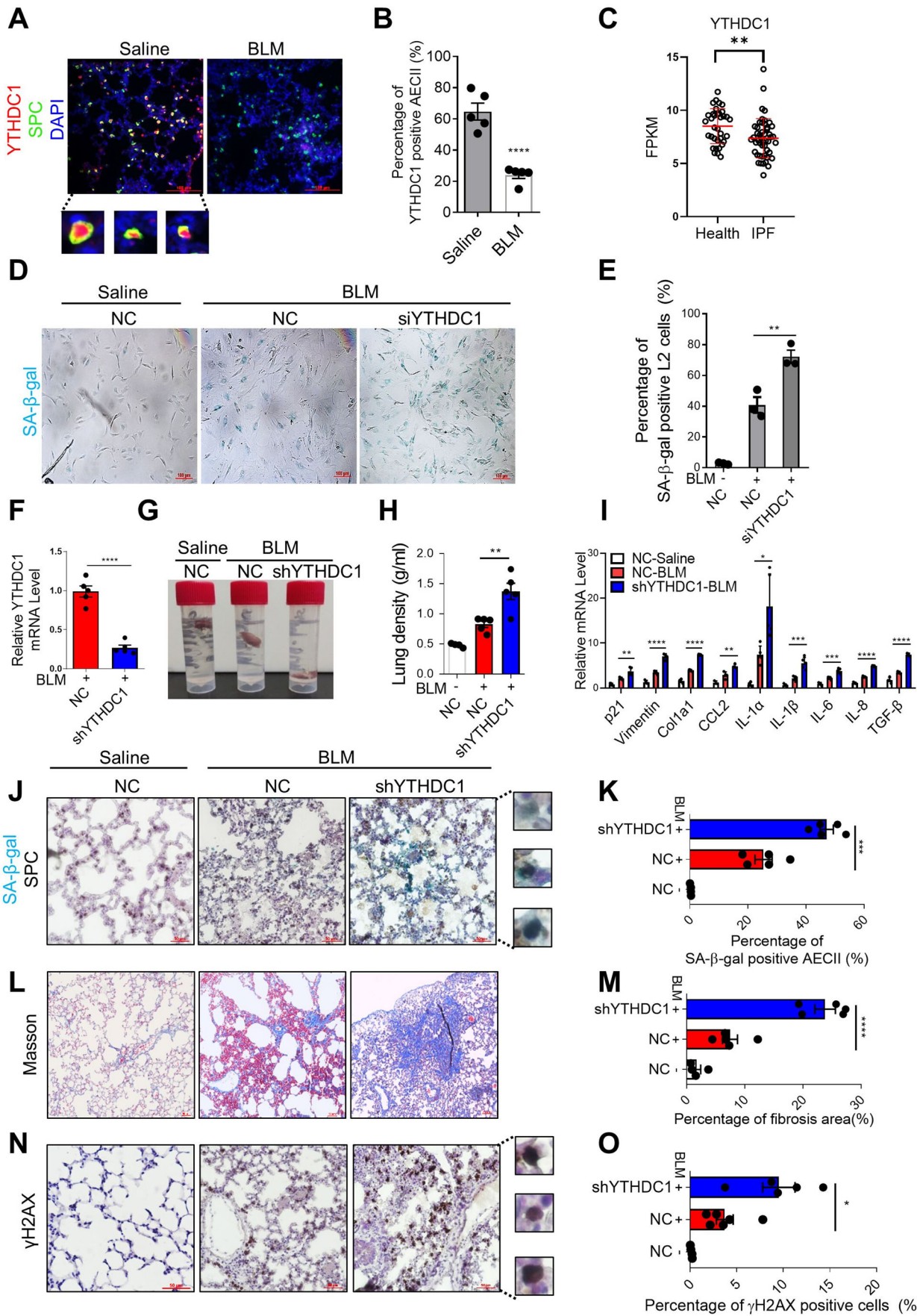

Figure 1. Downregulation of YTHDC1 exacerbates bleomycin-induced pulmonary senescence and fibrosis.

(A) YTHDC1 localizes in SPC-positive cells in mice lungs. Immunofluorescence was performed to determine the localization of YTHDC1 (red) and SPC (green) in mice lungs that treated with saline or BLM for 7 days. Inset shows colocalized foci at high magnification. Scale bar: 100 μm. (B) Quantification of panel A. The percentage of double-positive cells in YTHDC1-positive cells was calculated ($n = 5$ per group). (C) YTHDC1 mRNA decreases in IPF patients. YTHDC1 expression levels in normal (35 people) and IPF (49 people) human lung tissues were determined using published datasets (Data ref: Ahangari et al, 2019). (D) SA-β-gal staining of L2 cells transfected with NC or siYTHDC1. Forty-eight hours after transfection, cells were treated with BLM or saline for 4 days and subjected to SA-β-gal staining. Scale bars: 100 μm. (E) Quantification of D. The percentage of SA-β-gal positive cells was calculated ($n \geq 100$ cells × three repeats). (F) YTHDC1 expression level in mice lungs. C57/BL6 mice transfected with indicated shAAV vectors were treated with BLM for 7 days ($n = 5$ per group). YTHDC1 mRNA was analyzed by RT-qPCR. (G) Representative images of the position of mice lung tissues. C57/BL6 mice transfected with indicated shAAV vectors were treated with saline or BLM for 7 days. ($n \geq 4$ per group). The higher the density of lung tissue, the closer it is to the bottom of the tube. (H) Quantification of the density of lung (g/ml) from panel G. (I) SASP factors in mice lungs from panel G were detected by RT-qPCR. ($n \geq 4$ per group). (J) Representative images of SPC IHC with SA-β-gal staining in the mice lungs from panel G. ($n \geq 4$ per group). Inset shows colocalized foci at high magnification. Scale bar: 50 μm. (K) Quantification of panel J. The percentage of double-positive cells in SPC-positive cells was calculated. (L) Masson's trichrome was performed to determine the level of fibrosis from the mice lungs of panel G. ($n \geq 4$ per group). Scale bar: 50 μm. (M) Quantification of panel L. The percentage of fibrosis area (blue) was calculated. (N) Representative images of γH2AX IHC from the mice lungs of panel G. ($n \geq 4$ per group). Inset shows positive signals at high magnification. Scale bar: 50 μm. (O) Quantification of panel N. The percentage of γH2AX positive cells was calculated. Data information: All values are mean ± SEM. The unpaired Student's two-tailed $t$-test was used to determine the statistical significance (*$P < 0.05$, **$P < 0.01$, ***$P < 0.001$, ****$P < 0.0001$). $n =$ number of biological replicates. Source data are available online for this figure.

alleviated the senescence of AECII cell, but has no effect on other types of cells (Fig. EV2H–M). Altogether, these observations support the idea that YTHDC1 plays a protective role in stress-induced pulmonary cell senescence and fibrosis independent of its m6A recognition.

## YTHDC1 regulates ATR activation independent of m6A recognition

The progressive accumulation of DNA damage is thought to be one of the driving forces that initiate AECII senescence (Richeldi et al, 2017; Schuliga et al, 2021). Ataxia telangiectasia mutated (ATM) and ataxia telangiectasia-mutated and RAD3-related (ATR) act as critical master regulators of DDR signaling to promote DNA damage repair, and delay cell cycle progression until damages been repaired. Phosphorylation of ATM and ATR is an indispensable step for activation of downstream effectors in DDR. We wondered if the phosphorylation of ATM/ATR is regulated by YTHDC1. To answer this question, phosphorylated-ATM/ATR (p-ATM/p-ATR) are detected in YTHDC1-depleted A549 cells treating with VP-16. Less p-ATR and more p-ATM are observed in YTHDC1-depleted cells with no significant change of total ATM and ATR (Fig. 3A–C). It is reported that ATR is activated in an ATM/MRE11 dependent pathway in response to DSBs (Jazayeri et al, 2006; Myers and Cortez, 2006; Rodier et al, 2009), implying that YTHDC1 deficiency impairs ATM-dependent ATR activation (Fig. 3A). Given that YTHDC1 has been reported to participate DNA damage repair depending on its m6A binding activity (Zhang et al, 2020), we examined whether the m6A modification involves in the activation of ATR by knockdown several members of m6A readers, methyltransferases and demethyltransferases in VP-16 treated A549 cells. It reveals that only YTHDC1 knockdown decreases the ATR phosphorylation, whereas other m6A modification-related proteins have no effect (Figs. 3D–L and EV3A). This experiment has been repeated in BLM-treated A549 cells and got the same results (Fig. EV3B–J). Consistently, ectopic expression of either YTHDC1-WT or YTHDC1-MUT in YTHDC1-depleted A549 cells increases the phosphorylation of ATR (Fig. 3M-O). Altogether, there results suggest that YTHDC1 regulates ATR phosphorylation independent on its m6A reader function.

The coming question is whether the effects of YTHDC1 on stress-induced senescence and pulmonary fibrosis depends on the activation of ATR. To address this question, we detected the p-ATR in pulmonary fibrosis mice model and found that both the p-ATR foci number and the percentage of cells with p-ATR foci increase significantly in lung fibrosis and decrease back to normal levels in shYTHDC1 group (Fig. 3P–R).

## YTHDC1 regulates recruitment of TopBP1 to DSB sites

When DNA damage occur, ATR is recruited to RPA-coated ssDNA with the help of the ATR Interacting Protein (ATRIP) and activated by DNA topoisomerase 2-binding protein 1 (TopBP1) (Cortez et al, 2001). Simultaneously, TopBP1 can be recruited to the DNA damage sites mainly in two manners, with one depending on RAD9-Hus1-RAD1 (9-1-1) complex and RAD17 protein, and the other depending on MRE11-RAD50-NBS1 (MRN) complex (Duursma et al, 2013; Mooser et al, 2020). To explore the mechanism by which YTHDC1 regulates ATR activation, we determined whether YTHDC1 affects the recruitment of ATR activation regulators (TopBP1, ATRIP, RAD17, RAD9A, and RPA1) at damage sites (represent as γH2AX signal). As the results show, depletion of YTHDC1 impairs the localization of TopBP1 to DNA damage sites but does not affect the recruitment of ATRIP, RAD17, RAD9A or RPA1 (Figs. 4A–D and EV4A,B). Similarly, both TopBP1 nuclear foci and colocalized to γH2AX foci are decreased after YTHDC1 depleted in VP-16 treated cells, whereas the expression level of TopBP1 does not change (Figs. 4E–G and EV4C). It suggests that YTHDC1 is required for localization of TopBP1 to DNA damage sites. We also examined the TopBP1 foci in the YTHDC1 depleted A549 cells treated with BLM and got the same results (Fig. EV4D,E). As one of the TopBP1 recruitment mechanism is mediated by 9-1-1 complex and RAD17 protein, we wondered whether YTHDC1 functions together with 9-1-1 complex or RAD17 to recruit TopBP1. To answer this question, we knocked down YTHDC1, RAD9A, RAD17 respectively or concurrently and detected the TopBP1 foci in the nucleus after VP-16 treatment. The result displays that YTHDC1 knockdown together with RAD9A or RAD17 almost completely abolish the recruitment of TopBP1 to DSB sites whereas the single knockdown of any one only decreases the TopBP1 foci moderately (Fig. EV4F,G). In addition, the activation of ATR is also inhibited more significantly in YTHDC1 and RAD9A/RAD17 concurrent

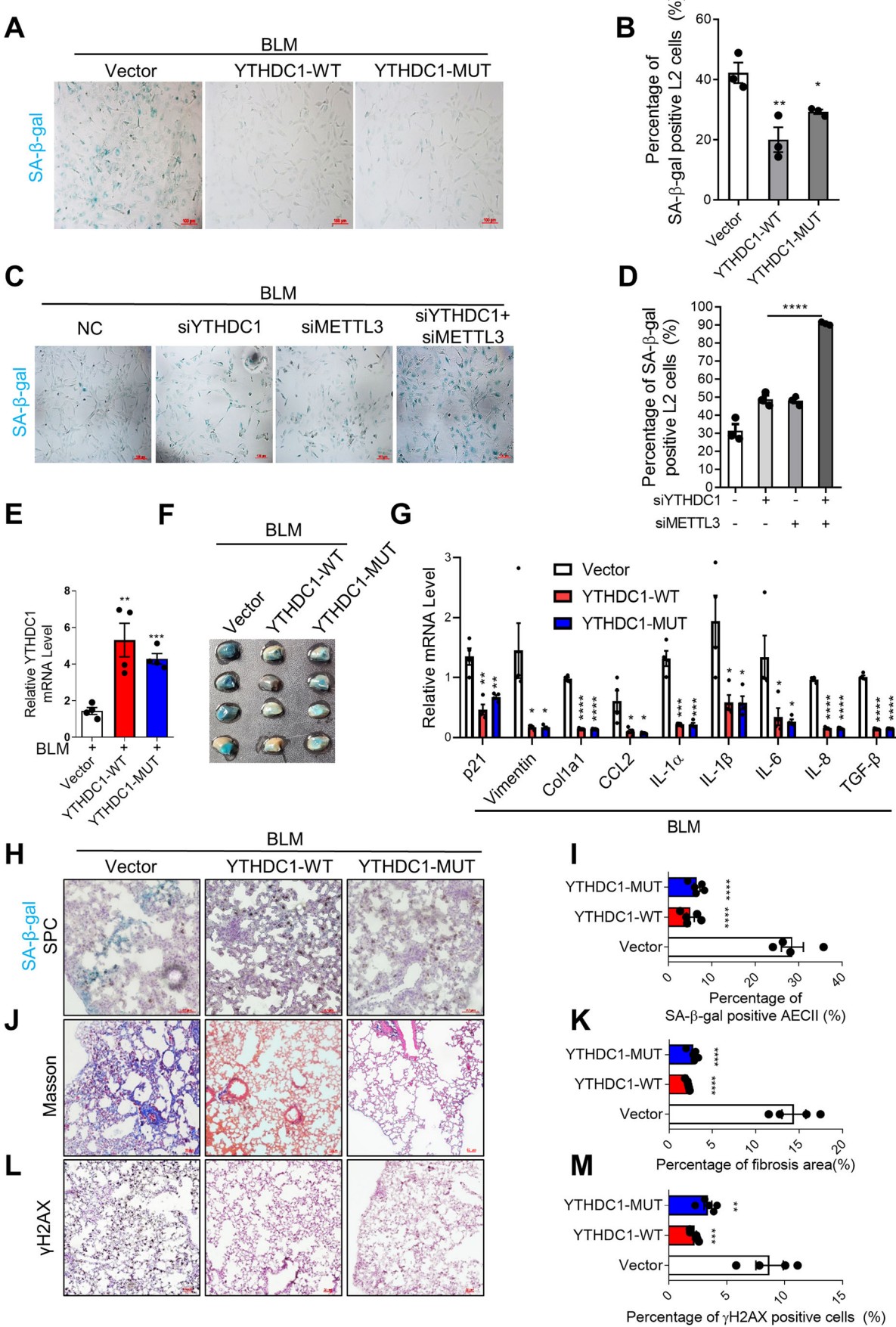

**Figure 2. Overexpression of YTHDC1 attenuates bleomycin-induced pulmonary senescence and fibrosis.**

(A) SA-β-gal staining of L2 cells overexpressed with Vector, YTHDC1-WT or YTHDC1-MUT. Forty-eight hours after transfection, cells were treated with BLM for 4 days and subjected to SA-β-gal staining. Scale bars: 100 μm. (B) Quantification of **A**. The percentage of SA-β-gal positive cells was calculated (*n* ≥ 100 cells × three repeats). (C) SA-β-gal staining of L2 cells transfected with indicated siRNAs. Forty-eight hours after transfection, cells were treated with BLM for 4 days and subjected to SA-β-gal staining. Scale bars: 100 μm. (D) Quantification of **C**. The percentage of SA-β-gal positive cells was calculated (*n* ≥ 100 cells × three repeats). (E) YTHDC1 expression level in mice lungs. C57/BL6 mice transfected with indicated AAV vectors were treated with BLM for 7 days (*n* ≥ 4 per group). YTHDC1 mRNA was analyzed by RT-qPCR. (F) Lungs from mice in panel **E** were used for SA-β-gal staining. (*n* ≥ 4 per group). (G) SASP factors were detected by RT-qPCR using mice lungs from panel **E**. (*n* ≥ 4 per group). (H) Representative images of SPC IHC with SA-β-gal staining in the mice lungs from panel **E**. (*n* ≥ 4 per group). Scale bar: 50 μm. (I) Quantification of panel **H**. The percentage of double-positive cells in SPC-positive cells was calculated. (J) Masson's trichrome was performed to determine the level of fibrosis in the mice lungs from panel **E**. (*n* ≥ 4 per group). Scale bar: 100 μm. (K) Quantification of panel **J**. The percentage of fibrosis area (blue) was quantified. (L) Representative images of γH2AX IHC in the mice lungs from panel **E**. (*n* ≥ 4 per group). Scale bar: 50 μm. (M) Quantification of panel **L**. The percentage of γH2AX positive cells was calculated. Data information: All values are mean ± SEM. The One-way ANOVA was used to determine the statistical significance (*$P < 0.05$, **$P < 0.01$, ***$P < 0.001$, ****$P < 0.0001$). *n* = number of biological replicates. Source data are available online for this figure.

knockdown groups than in individual knockdown groups, whereas simultaneous knockdown of RAD9A and RAD17 reduced p-ATR to a degree similar to knockdown of RAD9A or RAD17 alone (Fig. EV4H). These results supported the conclusion that YTHDC1 and the 9-1-1 complex work in different pathways to recruit TopBP1 and activate ATR.

Then we examined whether YTHDC1 regulates the recruitment of TopBP1 through affecting the other way that mediates by MRN complex (Duursma et al, 2013). The MRN complex recruitment is not affected by YTHDC1 depletion as the MRE11 foci is unchanged (Fig. EV5A,B), however, the precipitated MRE11 by TopBP1 IP decreased (Figs. 4H and EV5C), suggesting that knockdown of YTHDC1 impairs interaction between TopBP1 and MRE11. In addition, concurrent depletion of YTHDC1 and MRE11 inhibits the activity of ATR to a degree equal to knockdown of YTHDC1 or MRE11 respectively (Fig. 4I), suggesting that YTHDC1 regulates TopBP1 recruitment to the DNA damage sites through MRN complex.

Next, we explored whether YTHDC1 mediates TopBP1 recruitment is dependent on its m6A binding function. Similar to the phosphorylation of ATR, either overexpression of YTHDC1-WT or YTHDC1-MUT rescues the TopBP1 foci reduced by YTHDC1 depletion (Fig. EV5D–F). Altogether, these results suggest that YTHDC1 regulates the activation of ATR through promoting the recruitment of TopBP1 to DNA damage sites independent of m6A modification.

## YTHDC1 binds to TopBP1 directly

To address whether YTHDC1 directly binds to TopBP1, we performed co-immunoprecipitation (co-IP) assay in HEK293T cells. It reveals that Flag-YTHDC1 pulls down endogenous TopBP1 successfully in VP-16 or BLM-treated cells (Figs. 5A and EV5G). Interestingly, the endogenous MRE11 also been pulled down (Fig. 5A), indicating that YTHDC1 forms triplex with TopBP1 and MRE11. The reverse co-IP shows that Flag-TopBP1 pulls down endogenous YTHDC1 in VP-16 treated cells (Fig. 5B), suggesting the interaction of YTHDC1 and TopBP1 is influenced by DNA damage. We also found that the YTHDC1-MUT can bind to the TopBP1 the same as YTHDC1-WT, and the interaction is not disrupted by RNase A or Benzonase (BN) treatment (Fig. 5C), suggesting that the interaction between YTHDC1 and TopBP1 is not dependent on DNA or m6A modified RNA. To confirm YTHDC1 directly interacts with TopBP1, we expressed and purified mCherry-YTHDC1 and His-TopBP1 from insect SF9 cells and pull-down assay was performed in vitro. The result reveals that YTHDC1 can directly bind with TopBP1 in the absence of DNA and RNA (Fig. 5D). We further characterized the structure-function

relationship between YTHDC1 and TopBP1 by constructing truncated proteins of both of them. According to co-IP experiments in cells expressing Flag-labeled YTHDC1 fragments (NTD, YTH, or CTD domain), TopBP1 is pulled down by the NTD fragment but not the YTH or CTD fragment of YTHDC1 (Hazra et al, 2019) (Fig. 5E). More importantly, MRE11 also can precipitate with the NTD fragment (Fig. 5E), which may explain the results that the NTD domain is enough to recruit the TopBP1 to DNA damage sites and promote ATR activation (Fig. EV5H–L). In addition, we used truncated TopBP1 to perform the co-IP and found that Δ7–8 and 1-3 domain, but not Δ1–2 or 1–2 domain of TopBP1 can interact with YTHDC1, suggesting that BRCT domains (1, 2, and 3) of TopBP1 contributes to the interaction with YTHDC1 (Fig. 5F).

## Loss of YTHDC1 causes genomic instability

It is well recognized that failure of ATR activation will weaken the DNA damage repair, resulting in accumulation of DNA damage and gene instability (Hernandez-Segura et al, 2018; Lopez-Otin et al, 2013). We observed that γH2AX increases in YTHDC1 depleted A549 cells (Fig. 6A). The result of comet assay displays that there are more DNA fragments in YTHDC1 deficient cells, however, rescues by overexpression of either YTHDC1-WT or YTHDC1-MUT (Fig. 6B). Depletion of YTHDC1 also increases the level of micronuclei, indicating the deficiency in DNA damage repair and induction of genome instability (Fig. 6C,D). These results imply that YTHDC1 can participate the DNA damage repair to maintain the stability of genomics.

To this end, we tested whether YTHDC1 resists stress-induced cellular senescence through TopBP1-ATR pathway. We overexpressed the NTD domain of YTHDC1 in YTHDC1-depleted cells and found that the NTD domain, which is able to rescue ATR activation (Fig. EV5J), successfully rescues cellular senescence in YTHDC1 deficient cells (Fig. 6E–G). Moreover, depletion of TopBP1 counteracts the protective effect of YTHDC1 on senescence as shown by SA-β-gal staining and p21 expression level (Fig. 6H–J). Collectively, these results reveal that YTHDC1 plays a protective role in stress-induced senescence through regulating ATR activation by binding to TopBP1 via NTD domain.

## Discussion

Accumulating DNA damages is a leading factor of aging and aging related diseases. However, how the DNA lesions mount up is an

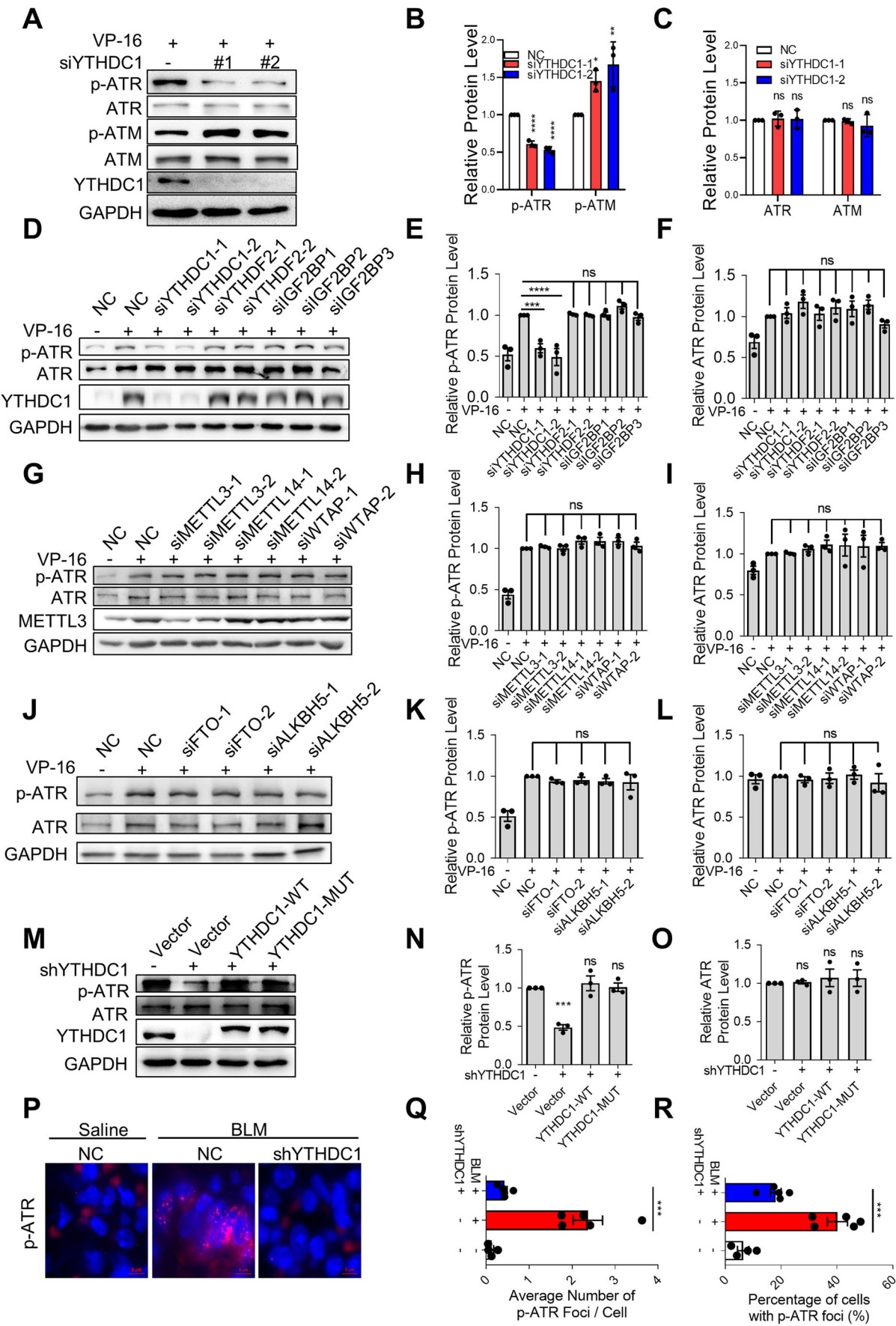

◀ **Figure 3. YTHDC1 is responsible for ATR activation.**

(A) Immunoblot analysis of activated ATR, total ATR, activated ATM, total ATM and YTHDC1 in A549 cells transfected with NC, siYTHDC1-1 or siYTHDC1-2. A549 cells were treated with VP-16 for 24 h prior to analysis. (B,C) Quantification of panel **A**. The relative protein p-ATR, p-ATM, ATR or ATM levels were determined by normalizing the intensities of p-ATR, p-ATM, ATR or ATM to the intensity of GAPDH. ($n = 3$). (D) Immunoblot analysis of activated ATR, total ATR, and YTHDC1 in A549 cells transfected with indicated siRNAs. A549 cells were treated with VP-16 or DMSO for 24 h prior to analysis. (E,F) Quantification of panel **D**. The relative protein p-ATR or ATR levels were determined by normalizing the intensities of p-ATR or ATR to the intensity of GAPDH. ($n = 3$). (G) Immunoblot analysis of activated ATR, total ATR, and METTL3 in A549 cells transfected with indicated siRNAs. A549 cells were treated with VP-16 or DMSO for 24 h prior to analysis. (H,I) Quantification of panel **G**. The relative protein p-ATR or ATR levels were determined by normalizing the intensities of p-ATR or ATR to the intensity of GAPDH. ($n = 3$). (J) Immunoblot analysis of activated ATR and total ATR in A549 cells transfected with indicated siRNAs. A549 cells were treated with VP-16 or DMSO for 24 h prior to analysis. (K,L) Quantification of panel **J**. The relative protein p-ATR or ATR levels were determined by normalizing the intensities of p-ATR or ATR to the intensity of GAPDH. ($n = 3$). (M) Immunoblot analysis of activated ATR, total ATR, and YTHDC1 in YTHDC1-deficient A549 cells overexpressed with Vector, YTHDC1-WT or YTHDC1-MUT. Cells were treated with VP-16 for 24 h prior to analysis. (N,O) Quantification of panel **M**. The relative protein p-ATR or ATR levels were determined by normalizing the intensities of p-ATR or ATR to the intensity of GAPDH. ($n = 3$). (P) Immunofluorescence (IF) detection of p-ATR foci in the lungs of C57/BL6 mice from NC or shYTHDC1 groups treated with BLM or saline ($n \geq 4$ mice per group). Scale bar: 5 μm. (Q) Quantification of panel **P**. The average number of p-ATR foci per cell. ($n \geq 4$ per group). (R) Quantification of panel **P**. The percentage of cells with p-ATR foci was calculated ($n \geq 4$ per group). Data information: All values are mean ± SEM. The unpaired Student's two-tailed $t$-test was used to determine the statistical significance between two groups. The One-way ANOVA was used to determine the statistical significance for more than two groups (*$P < 0.05$, **$P < 0.01$, ***$P < 0.001$, ****$P < 0.0001$. $n$ = number of biological replicates. Source data are available online for this figure.

important and attractive question. In this study, we found that loss of YTHDC1 expression is probably one of the answers. YTHDC1 is a well-known m6A binding protein that regulating RNA splicing, stability et al (Widagdo et al, 2022). Here, we identified a noncanonical, m6A binding-independent role of YTHDC1 in maintaining genomic stability and protecting lung against stress-induced senescence and fibrosis. In detail, in response to DNA lesion, YTHDC1 forms complexes with TopBP1, regulates TopBP1 interacting with MRE11 and then activates ATR. We observed reduced YTHDC1 expression in both IPF patients and pulmonary fibrosis mice model. Hence, it is possible that decreased expression of YTHDC1 impairs ATR activation and DNA damage repair, leading to cellular senescence and diseases, such as idiopathic pulmonary fibrosis. Our findings expand the biological functions of YTHDC1 to an ATR activation regulator, and provide a potential strategy for improving DNA damage repair ability via restoration of YTHDC1 expression.

## YTHDC1 is a suppresser of pulmonary senescence and fibrosis

Idiopathic pulmonary fibrosis is a fatal disease with unclear and complicated mechanism, which needs exploring. In this study, we found that YTHDC1 plays a protective role during IPF development, which probably provide a clue for a new therapy solution. Firstly, we observed that YTHDC1 primarily expresses in pulmonary alveolar epithelial type 2 cells. However, its expression level is significantly reduced in lung tissues from pulmonary fibrosis mice and IPF patients. The reduction of YTHDC1 probably leads to pulmonary fibrosis progress, as we observed that knockdown of YTHDC1 exacerbates stress-induced senescence and pulmonary fibrosis. We also overexpressed YTHDC1 in mice lung tissue and in cultured AECII cell line. Excitingly, the results revealed that overexpression of YTHDC1 prevents senescence both in vivo and in vitro, and further antagonizes pulmonary fibrosis. Although we observed the anti-aging function of YTHDC1 in AECII, we cannot rule out the effect of YTHDC1 on other lung cell types in lung fibrosis. In spite of this, these results still indicate YTHDC1 is an important regulator in senescence and provides a potential therapeutic target in reversing

cellular senescence and treating age-related disorders such as pulmonary fibrosis.

## YTHDC1 antagonizes cell senescence and pulmonary fibrosis by regulating the activation of ATR

Deficiencies in DNA repair mechanisms cause accelerated aging and underlie several human progeroid syndromes such as Seckel syndrome, which characterized by severe deficiency in ATR and high levels of DNA damage (Chen et al, 2007; Murga et al, 2009; Yousefzadeh et al, 2021). Similarly, we found that the cellular mechanism by which YTHDC1 mediates stress-induced pulmonary senescence and fibrosis is that YTHDC1 deficiency impairs ATR activation, resulting in the accumulation of unrepaired DNA damage and genomic instability. In addition, overexpression of the NTD domain of YTHDC1, which is able to rescue ATR activation, successfully alleviates cellular senescence induced by depletion of YTHDC1. In addition, depletion of TopBP1 counteracts the protective effect of YTHDC1 on senescence. These observations are consistent with the previous finding that ATR mutation accelerates senescence in mice and that either ATR null or YTHDC1 null mice are enviable (Cimprich and Cortez, 2008; Kasowitz et al, 2018). Hence, these findings support the idea that YTHDC1 acts in the ATR pathway to antagonize stress-induced senescence.

Our previous study discovered that YTHDC1 promotes homologous recombination-mediated DNA damage repair by protecting m6A-modified RNAs and thus facilitate recruitment of the damage repair factor such as RAD51, BRCA1 to DSB sites (Zhang et al, 2020). Here, we found the YTHDC1 mainly regulates the localization of TopBP1 to damage sites to participate the activation of ATR. These two findings proved that YTHDC1 plays two different roles in DNA damage response and repair. One is to regulate ATR activation during DNA damage response, the other is to promote damage repair by help recruiting repair factors. The dual roles of YTHDC1 in DDR is similar to BRCA1, which involves in both DNA end resection and RAD51 recruitment during homologous recombination-mediated DSB repair (Caestecker and Van de Walle, 2013; Shrivastav et al, 2008), revealing that YTHDC1 is key factor in DNA damage repair.

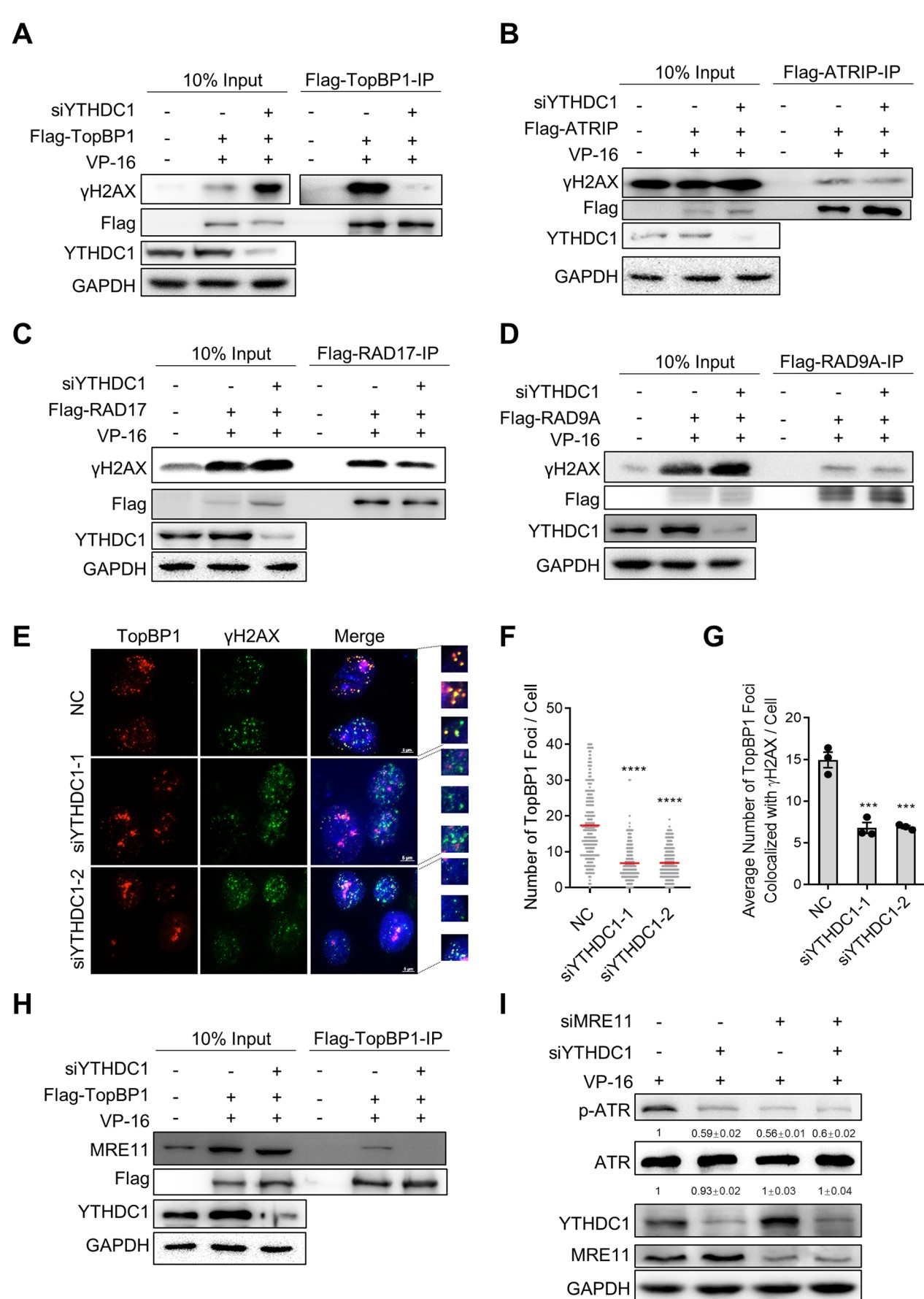

◄

**Figure 4. YTHDC1 regulates the recruitment of TopBP1 to MRN complex.**

(A) Co-IP assay to determine the interaction of TopBP1 with γH2AX in HEK293T cells. YTHDC1-depleted cells were transfected with Flag-TopBP1 or Vector and treated with VP-16 for 24 h. Cell lysates were used for immunoprecipitation with Flag-beads. Immunoprecipitates were immunoblotted with γH2AX, Flag, YTHDC1, and GAPDH antibody, respectively. ($n = 3$). (B) The same as in A, except that Flag-ATRIP was over-expressed in HEK293T cells. ($n = 3$). (C) The same as in A, except that Flag-RAD17 was over-expressed in HEK293T cells. ($n = 3$). (D) The same as in A, except that Flag-RAD9A was over-expressed in HEK293T cells. ($n = 3$). (E) Immunofluorescence (IF) detection of TopBP1 and γH2AX foci in control or YTHDC1-depleted A549 cells. Cells were treated with VP-16 for 24 h before detection. Scale bar: 5 μm. (F) Quantification of E. The number of TopBP1 foci per cell ($n \geq 100$ cells × three repeats). (G) Quantification of E. The average number of TopBP1 foci colocalized with γH2AX foci per cell ($n \geq 100$ cells × three repeats). All values are the average ±SEM of three independent experiments. The One-way ANOVA was used to determine the statistical significance (**$P < 0.01$, ****$P < 0.0001$). (H) Co-IP assay to determine the interaction of TopBP1 with MRE11 in HEK293T cells. YTHDC1-depleted cells were transfected with Flag-TopBP1 or Vector and treated with VP-16 for 24 h. Cell lysates were used for immunoprecipitation with Flag-beads. Immunoprecipitates were immunoblotted with MRE11, Flag, YTHDC1, and GAPDH antibody, respectively. ($n = 3$). (I) Immunoblot analysis of activated ATR, total ATR, YTHDC1, and MRE11 in A549 cells transfected with NC, siYTHDC1, siMRE11 or siYTHDC1 and siMRE11. Cells were treated with VP-16 for 24 h prior to analysis. ($n = 3$). Data information: All values are mean ± SEM. The One-way ANOVA was used to determine the statistical significance (***$P < 0.001$, ****$P < 0.0001$). $n =$ number of biological replicates. Source data are available online for this figure.

## YTHDC1 activates ATR by mediating localization of TopBP1 to DSB sites

Upon replication stress occurs, TopBP1 is recruited to ssDNA-dsDNA junction via the 9-1-1 complex to activate ATR (Kumagai et al, 2006). Whereas in the presence of DSBs, TopBP1 can also be recruited by MRN complex to activate ATR (Duursma et al, 2013; Mooser et al, 2020; Myers and Cortez, 2006). Here, we proposed that YTHDC1 directly interacts with TopBP1 and is required for its recruitment to MRN complex to promote ATR activation, which is independent of 9-1-1 complex, as suggested by the following observations. First, depletion of YTHDC1 only affects the localization of TopBP1 to DSB sites but no other ATR activation regulators, such as ATRIP, RAD9A or RAD17. Second, in vivo and in vitro co-IP assays show that YTHDC1 directly interacts with TopBP1. Third, YTHDC1 immunoprecipitation find that YTHDC1 also binds to MRE11. Knockdown of YTHDC1 doesn't affect the localization of MRE11 at DNA damage sites but weakens the interaction between MRE11 and TopBP1. It seems that YTHDC1 functions as a scaffold factor that regulating the recruitment of TopBP1 by MRE11. Fourth, the effect of YTHDC1 on resistant to stress-induced senescence are impaired by the depletion of TopBP1, suggesting that YTHDC1 and TopBP1 act in same pathway to resistant stress-induced senescence. Furthermore, the process of YTHDC1 regulating the interaction between TopBP1 and MRE11 to affect the activation of ATR is consistent with the existing model that ATM and MRE11 are required for ATR to receive DSBs signals (Jazayeri et al, 2006; Myers and Cortez, 2006; Saha et al, 2013). Therefore, knockdown of YTHDC1 prevents DNA damage signals from being transmitted from ATM to ATR, inhibits ATR activation, and leads to persistent DNA damage.

## The new function of YTHDC1 is independent of m6A

YTHDC1 is a well-known m6A reader protein and has been heavily studied in the last few decades (Xu et al, 2015). Here, we uncovered a noncanonical, m6A binding-independent role of YTHDC1 in regulating the activation of ATR and protecting lung against stress-induced senescence. We found that knockdown of any one of m6A methylases, demethylases or m6A reader proteins other than YTHDC1 do not influence the activation of ATR, whereas knockdown of YTHDC1 significantly decreases the phosphorylation of ATR. Furthermore, YTHDC1 mutation that lacking m6A binding activity is able to promote TopBP1 recruitment, ATR activation and then attenuate cell senescence the same as wild-type YTHDC1. Similarly, other members involve in m6A modification have also been reported to display m6A-independent functions. For examples, METTL3 and METTL14 complex promotes SASP transcription during senescence, the YTHDC2 is essential for mammalian fertility, and METTL16 facilitates translation in cytosol (Liu et al, 2021; Li et al, 2022; Su et al, 2022). Altogether, these findings prove that many of the m6A-related proteins exert both m6A-dependent and m6A-independent functions.

## Methods

### Cell culture

HEK293T, A549, and L2 cells were obtained from American Type Culture Collection (Manassas, VA). HEK293T and A549 cells were grown in DMEM (GIBCO) with 10% fetal bovine serum (GIBCO) and 100 U/ml penicillin/streptomycin (GIBCO). L2 Cells were grown in F-12K Medium (ATCC, 30-2004) with 10% fetal bovine serum (GIBCO) and 100 U/ml penicillin/streptomycin (GIBCO). All cells were cultured at 37 °C and 5% $CO_2$. All cells were negative of mycoplasma contamination.

### Plasmids and transfection

Human wild-type YTHDC1, TopBP1, RDA9A, RAD17, and ATRIP genes were amplified from MCR5 mRNA and cloned into pLenti-HA/Flag. The siRNA-resistant YTHDC1 was generated by synonymously mutating the sequences at siRNA targeting site based on wild-type YTHDC1. Mouse wild-type YTHDC1 was amplified from mouse lung mRNA and cloned into pAAV-MCS. Rat YTHDC1 was amplified from L2 cells mRNA and cloned into pLVX-AcGFP1-N1. The m6A binding site mutated YTHDC1 (YTHDC1-MUT, referred to as W378A and W428A) was generated by site-directed mutation based on wild-type YTHDC1.

Plasmid were transfected into indicated cells using Lipo3000 following the manufacturer's instruction (Thermo Fisher Scientific). siRNAs were transfected into indicated cells using Lipofectamine RNAiMAX reagent (Thermo Fisher Scientific). siRNA sequences are listed in Table EV1.

### Cell treatment

Unless otherwise indicated, cell lines were treated with Etoposide (VP-16, 10 μM, MCE) for 24 h to induce DNA damage and 100 μM for 24 h, then released 7 days to induce senescence; Bleomycin (10 μg/ml, Selleck) for 4 h to induce DNA damage and 20 μg/ml 4 days to induce senescence.

## Immunofluorescence

For cell immunofluorescence experiments, cells plated on coverslips were fixed in cold methanol and permeabilized with 0.2% Triton. The coverslips were incubated sequentially with primary antibody and fluorescence-labeled secondary antibody. Coverslips mounted with Vectashield mounting medium containing DAPI (Vector Laboratories) were visualized and analyzed using fluorescence microscopy. Antibodies: RPA1 (1:100, sc-28304, Santa Cruz), TopBP1 (1:100, sc-271043, Santa Cruz), MRE11 (1:100,

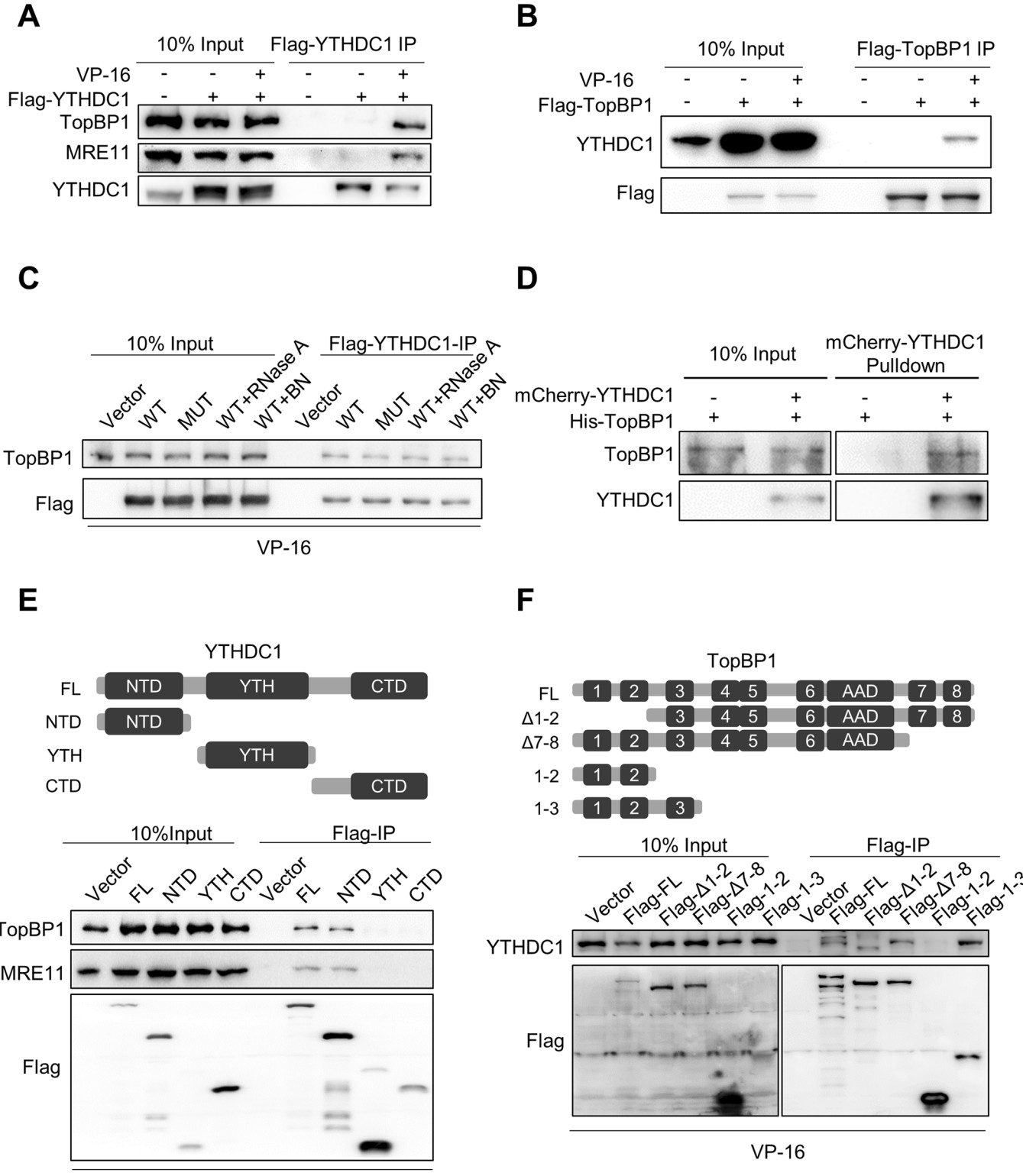

**Figure 5. YTHDC1 directly interacts with ToPBP1.**

(A) Co-IP assay to determine the interaction between YTHDC1 and TopBP1 or MRE11. HEK293T cells transfected with Flag-YTHDC1 were treated with VP-16 or DMSO for 24 h. Cell lysates were used for immunoprecipitation with Flag-beads. Immunoprecipitates were immunoblotted with TopBP1, MRE11, and YTHDC1 antibody, respectively. ($n = 3$). (B) Co-IP assay to determine the interaction between TopBP1 and YTHDC1. HEK293T cells transfected with Flag-TopBP1 were treated with VP-16 or DMSO for 24 h. Cell lysates were used for immunoprecipitation with Flag-beads. Immunoprecipitates were immunoblotted with YTHDC1 and Flag antibody, respectively. ($n = 3$). (C) The interaction between YTHDC1 and TopBP1 is independent of nucleic acids. HEK293T cells transfected with Flag-YTHDC1-WT or Flag-YTHDC1-MUT were treated with VP-16 for 24 h. Cell lysates were treated with RNase A or Benzonase prior to IP. Immunoprecipitates were immunoblotted with TopBP1 and Flag antibody, respectively. ($n = 3$). (D) YTHDC1 binds TopBP1 directly. In vitro pull-down assay using exogenously purified mCherry-YTHDC1 protein and His-TopBP1 protein. mCherry protein was added in the control group. ($n = 3$). (E) Up: Schematic diagram showing the domains of YTHDC1. Down: Co-IP assay to determine the domain of YTHDC1 that interacts with TopBP1 and MRE11. HEK293T cells transfected with Flag-YTHDC1 or its domain (NTD, YTH, and CTD) were treated with VP-16 for 24 h. Cell lysates were used for immunoprecipitation with Flag-beads. Immunoprecipitates were immunoblotted with TopBP1, MRE11, and Flag antibody, respectively. ($n = 3$). (F) Up: Schematic diagram showing the domains of TopBP1. Down: Co-IP assay to determine the domain of TopBP1 that interacts with YTHDC1. HEK293T cells transfected with Flag-TopBP1 or its truncated proteins were treated with VP-16 for 24 h. Cell lysates were used for immunoprecipitation with Flag-beads. Immunoprecipitates were immunoblotted with YTHDC1 and Flag antibody, respectively. ($n = 3$). Data information: $n =$ number of biological replicates. Source data are available online for this figure.

GTX70212, GeneTex). Lung immunofluorescence experiments was performed in lung sections stained with indicated protein antibody using Tyramide signal amplification kit (Servicebio, G1236) according to the standard procedures. Antibodies: YTHDC1(1:500, ab122340, Abcam), FTO (1:500, 27226-1-AP, Proteintech), WTAP (1:500, HPA01550, Sigma), SPC (1:500, ab211326, Abcam), Ki67 (1:400, ab16667, Abcam). Second antibody used: HRP-conjugated anti-rabbit (KPL, Inc).

## Co-immunoprecipitation

HEK293T cells were transfected with plasmids or indicated siRNA and treated with VP-16. Cells were lysed by RIPA buffer (1% NP-40, 0.25% Sodium Deoxycholate, 50 mM Tris-HCl (pH 7.4), 150 mM NaCl, 1 mM EDTA) and incubated with Flag antibody conjugated to protein G Dynabeads (MCE) overnight at 4 °C. Coprecipitating proteins were identified by SDS-PAGE followed by immunoblotting.

## Comet assay

Cells defected YTHDC1 were overexpressed YTHDC1-WT, YTHDC1-MUT or Vector and treated with VP-16. The cells were harvested and mixed with 0.5% low melting temperature agarose and layered on slides pre-coated by 1.5% normal agarose. Slides were lysed in 2.5 M NaCl, 100 mM EDTA, 10 mM Tris (pH 8.0), 0.5% Triton X-100, 3% DMSO, 1% N-lauroylsarcosine overnight at 4 °C and then electrophoresis in 300 mM sodium acetate, 100 mM Tris-HCl, 1% DMSO at 1.5 V/cm for 20 min. After neutralization with 0.4 M Tris-HCl (pH 7.3), slides were washed and dried with ethanol. The slides were then mounted with Vectashield mounting medium containing DAPI (Vector Laboratories) and visualized under fluorescence microscopy. Analysis was performed with CASP.

## Protein purification

YTHDC1 or TopBP1 protein expression was followed Bac-to-Bac Baculovirus Expression System. SF9 cells were collected by centrifuging at 1200RPM for 5 min and resuspended by lysis buffer (10 mM Tris-HCl (pH 8.0), 10 mM NaCl, 20 mM Imidazole) after highly expressed YTHDC1 or TopBP1. After disrupted cells by sonication (power 25%, pulse 2 s, off 15 s, 5 min),

YTHDC1 or TopBP1 protein was isolated by nickel affinity purification.

## In vitro pull-down assay

mCherry-YTHDC1 and His-TopBP1 proteins purified from SF9 cells were incubated with mCherry-Beads in RIPA buffer (1% NP-40, 0.25% sodium deoxycholate, 50 mmol/L Tris-HCl (pH 7.4) and 150 mmol/L NaCl) overnight at 4 °C, then washed four times with RIPA buffer. The samples were analyzed by immunoblotting using the following antibodies: YTHDC1 (1:2000, ab122340, Abcam), TopBP1 (1:1000, sc-271043, Santa Cruz).

## Mice treated with bleomycin and adenovirus

Animal use was approved by the ethical committee of Sun Yat-sen University (Guangzhou, China). C57BL/6 mice were purchased from Guangdong Medical Experimental Animal Center. 8-week-old mice were anaesthetized with isoflurane and injected intratracheally with specific shRNAs targeting YTHDC1 or control, or adenovirus overexpressing YTHDC1-WT, YTHDC1-MUT or Vector ($2 \times 10^{12}$ pfu). After 3 days, mice were anaesthetized with isoflurane and injected intratracheally with bleomycin or saline. 7 days after the inoculation, mice were sacrificed.

## SA-β-gal staining

The senescence-associated beta-galactosidase (SA-β-gal) staining assay was performed using an SA-β-gal staining kit (Beyotime, China) following the manufacturer's instructions.

## Cell isolation

AECII and fibroblasts cells were isolated from mice lungs treated with BLM or saline as previously described (Konigshoff et al, 2009; Konigshoff et al, 2007). In brief, lungs were lavaged with 20 ml sterile PBS and tissues minced and digested with 10 mg/ml dispase (Sigma-Aldrich, D4693), 20 mg/ml collagenase type I (Gibco, Thermo Fisher Scientific, 9001-12-1) and 0.02 g/L DNase I for 0.5 h at 37 °C. The suspension was sequentially filtered through 100-, 20-, and 10-µm nylon meshes and centrifuged at $200 \times g$ for 10 min. The cells were suspended with red cell lysate at room temperature for 5 min, centrifuged at $200 \times g$ for 5 min, and washed twice with PBS. The pellet was resuspended in DMEM, and negative selection for lymphocytes/

macrophages was performed by incubation on CD16/32- and CD45-coated Petri dishes for 30 min at 37 °C. Fibroblasts were isolated by adherence for 45 minutes on cell culture dishes coated with mouse IgG solution twice. And the remaining cells are AECII.

## Immunohistochemical analysis

Lung tissues were fixed overnight in 4% paraformaldehyde at room temperature, transferred to 70% ethanol and embedded in paraffin.

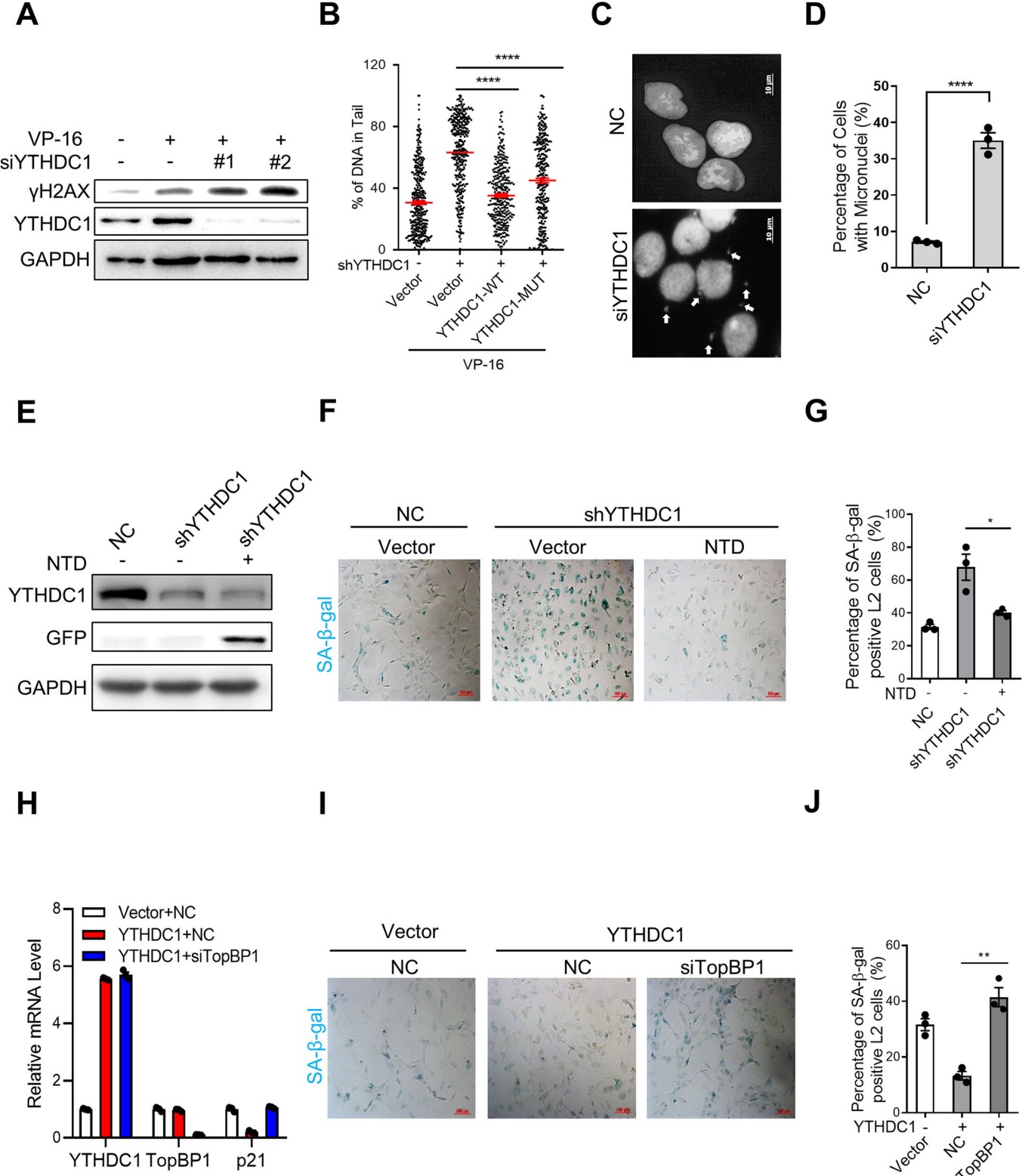

**Figure 6.   YTHDC1 attenuates stress-induced pulmonary epithelial cell senescence through TopBP1-ATR pathway.**

(A) Immunoblot analysis of γH2AX and YTHDC1 in A549 cells transfected with NC, siYTHDC1-1 or siYTHDC1-2. Cells were treated with VP-16 or DMSO for 24 h prior to analysis. (B) Quantification of the DNA tail in Comet assay. Comet assay detection of DNA fragments in normal and YTHDC1-deficient A549 cells overexpressed with Vector, YTHDC1-WT or YTHDC1-MUT. Cells were treated with VP-16 for 24 h prior to analysis. (C) Representative images showing micronuclei in normal and YTHDC1 depleted A549 cells. Cells were treated with VP-16 for 24 h before analysis. White arrowheads indicate the signal of micronuclei. Scale bar: 10 µm. (D) Quantification of C. The percentage of cells with micronuclei ($n \geq 100$ cells × three repeats). (E) Immunoblot analysis of YTHDC1 and GFP in YTHDC1-depleted L2 cells overexpressed with Vector or N terminal domain of YTHDC1 (GFP-NTD). Forty-eight hours after transfection, cells were treated with BLM for 4 days prior to analysis. (F) SA-β-gal staining of cells from panel E. Scale bars: 100 µm. (G) Quantification of F. The percentage of SA-β-gal positive cells was calculated ($n \geq 100$ cells × three repeats). (H) RT-qPCR analysis of the YTHDC1, TopBP1, and p21 mRNA levels in YTHDC1-overexpressed L2 cells transfected with NC or siTopBP1. Forty-eight hours after transfection, cells were treated with BLM for 4 days and subjected to RT-qPCR. (I) SA-β-gal staining of cells from panel H. Scale bars: 100 µm. (J) Quantification of I. The percentage of SA-β-gal positive cells was calculated ($n \geq 100$ cells × three repeats). Data information: All values are mean ± SEM. The unpaired Student's two-tailed $t$-test was used to determine the statistical significance between two groups. The One-way ANOVA was used to determine the statistical significance for more than two groups (*$P < 0.05$, **$P < 0.01$, ****$P < 0.0001$). $n$ = number of biological replicates. Source data are available online for this figure.

4 µm sections on slides were dewaxed in xylene, and then sequentially rehydrated in 100%, 95%, 70% ethanol and PBS buffer. Histopathological analysis of paraffin-embedded lungs was performed in lung sections stained with Masson trichrome (MXB Biotechnologies) using standard procedures. For immunohistochemistry, sections were blocked with 5% goat serum in PBS buffer, then incubated overnight with primary antibodies at 4 °C, washed three times with 1 × PBS, and incubated with secondary antibodies. Sections were washed three times with 1 × PBS and stained with DAB and hematoxylin (for DNA staining) according to standard protocols. Primary antibody used: γH2AX (1:400, 9718, CST). ATR pT1989 (1:100, GTX128145, GeneTex), YTHDC1 (1:400, ab122340, Abcam), p21 (1:1000, ab188224, Abcam), p16 (1:100, ab211542, Abcam), α-SMA (1:1000, 14395-1-AP, Proteintech). Second antibody used: HRP-conjugated anti-rabbit (KPL, Inc).

### Determination of lung density in mice

The lung density was measured by drainage method, which was as follows: (1) Mice were killed after anesthesia, and their lungs were taken and weighed and recorded as Mlung; (2) Then recorded the weight of 1.5 ml centrifuge tube without and full of saline calculate the exact volume of tube Vtube according to the weight and density of saline. (3) The lung tissue was completely submerged in the saline of the centrifuge tube, the air was removed, the lid was covered, and the water stains around the tube were wiped and recorded as Mtube; (4) Calculate the weight of saline in tube Msaline = Mtube-Mlung; (5) Calculate the volume of normal saline in the tube Vsaline = Msaline×ρsaline, ρsaline is the density of normal saline; (6) Calculate lung volume Vlung = Vtube-Vsaline; (7) calculated lung density ρlung = Mlung/Vlung.

### Immunoblotting

Proteins were separated with SDS-PAGE and transferred to PVDF membrane. The following antibodies for immunoblotting: YTHDC1 (1:2000, ab122340, Abcam), METTL3 (1:2000, ab195352, Abcam), ATR (1:5000, ab2905, Abcam), ATR pT1989 (1:1000, GTX128145, GeneTex), ATM pS1981 (1:5000, ab81292, Abcam), ATM (1:5000, ab32420, Abcam), γH2AX (1:2000, 9718, CST), TopBP1 (1:1000, sc-271043, Santa Cruz), RAD9A (1:2000, 13035-1-AP, Proteintech), RAD17 (1:1000, sc-17761, Santa Cruz), MRE11 (1:2000, GTX70212, GeneTex), GAPDH (1:5000, 60004-1-Ig, Proteintech), Flag (1:2000,

F1804, Sigma Aldrich), HRP-conjugated anti-rabbit or anti-mouse (KPL, Inc) were used as secondary antibody.

### RNA extraction and real-time quantitative RT-PCR

Total RNA was extracted using were extracted by Trizol extraction, and reverse-transcribed using PrimeScript RT reagent kit (AU311, TransGen Biotech), followed by amplification with RealStar Power SYBR Mixture (GenStar). qPCR was performed with a LightCycler 480 Real-Time PCR system (Roche). Data were analyzed using the comparative Ct (2-ΔΔCt) method. All experiments were performed in triplicate. qPCR primers for mRNA detection are listed in Table EV1.

### The characterization of IPF patients from GSE124685

RNAseq data were downloaded from Gene Expression Omnibus (GEO) database. Data ref (Ahangari et al, 2019). The patients used in the study were males and matched for age (control: 57.8 ± 10.7 & IPF: 57 ± 5.1), 59 samples were available from IPF patient lungs and 36 samples from healthy people. Samples were histologically assessed by a pathologist and 11 samples with emphysema, large airway, or large vessels were excluded from the analysis (McDonough et al, 2019).

### Statistical analysis

All experiments in this study have been performed at least three independent biological replicates. GraphPad Prism 8 was used for statistical analysis. Results are shown as mean ± SEM and the unpaired Student's two-tailed t-test was used to determine the statistical significance between two groups. The One-way ANOVA was used to determine the statistical significance for more than two groups (*$P < 0.05$; **$P < 0.01$; ***$P < 0.001$; ****$P < 0.0001$). For every Figure, statistical tests are justified as appropriate.

## Data availability

All data have been included in the manuscript, Figs, supplemental information and the Source Data files. This study includes no data deposited in external repositories.

## Peer review information

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

## Acknowledgements

We are grateful to members in Dr. Zhao's laboratory for insightful discussion. This work was supported by the National Natural Science Foundation of China Grants (82201734, 82171549, 31970683, 82101659); the Guangdong Basic and Applied Basic Research Foundation (2021A1515110989, 2021A1515010848), the Shenzhen Science and Technology Program (JCYJ20220530144214032).

## Author contributions

**Canfeng Zhang:** Data curation; Funding acquisition; Investigation; Writing—original draft; Writing—review and editing. **Liping Chen:** Investigation. **Chen Xie:** Investigation. **Fengwei Wang:** Investigation. **Juan Wang:** Investigation. **Haoxian Zhou:** Investigation. **Qianyi Liu:** Investigation. **Zhuo Zeng:** Investigation. **Na Li:** Investigation. **Junjiu Huang:** Conceptualization. **Yong Zhao:** Supervision; Project administration. **Haiying Liu:** Data curation; Supervision; Funding acquisition; Project administration; Writing—review and editing.

## Disclosure and competing interests statement

The authors declare no competing interests.

# Expanded View Figures

**Figure EV1.  Knockdown of YTHDC1 accelerates stress-induced pulmonary epithelial cell senescence.**

(A) METTL3, METTL14, WTAP, FTO and ALKBH5 expression levels in normal (35 people) and IPF (49 people) human lung tissues were determined using published datasets (Data ref: Ahangari et al, 2019). (B) YTHDF1, YTHDF2 and YTHDC2 expression levels in normal (35 people) and IPF (49 people) human lung tissues were determined using published datasets (Data ref: Ahangari et al, 2019). (C) Immunofluorescence was performed to determine the localization of WTAP/FTO (red) and SPC (green) in mice lungs that treated with saline or BLM for 7 days. Scale bar: 100 μm. (D) Quantification of panel C. The percentage of double-positive cells in WTAP/FTO-positive cells was calculated ($n = 5$ per group). (E) IF detection of Ki67 in control or YTHDC1-depleted L2 cells. Cells were treated with BLM for 4 days. Scale bar: 10 μm. (F) Quantification of E. The percentage of Ki67 positive cells was calculated ($n \geq 100$ cells × three repeats). (G) YTHDC1, p21, p16 and SASP factors were detected by RT-qPCR using L2 cells transfected with NC or siYTHDC1. Forty-eight hours after transfection, cells were treated with BLM or saline for 4 days. (H) SA-β-gal staining of L2 cells transfected with NC or siYTHDC1. Forty-eight hours after transfection, cells were treated with VP-16 for 1 days and released to 7 days and subjected to SA-β-gal staining. Scale bars: 100 μm. (I) Quantification of H. The percentage of SA-β-gal positive cells was calculated ($n \geq 100$ cells × three repeats). (J) Representative images of p21 in the mice lungs ($n \geq 4$ per group). C57/BL6 mice transfected with indicated AAV vectors were treated with BLM for 7 days ($n \geq 4$ per group). Inset shows positive signals at high magnification. from the mice lungs. Scale bar: 50μm. (K) Quantification of panel J. The percentage of p21 positive cells was calculated. (L, M), as in panels J,K, except using p16 antibody to perform the IHC. ($n \geq 4$ per group). Scale bar: 50 μm. (N,O) as in panels J,K, except using α-SMA antibody to perform the IHC. ($n \geq 4$ per group). Scale bar: 50 μm. (P) Immunofluorescence (IF) detection of γH2AX foci in the mice lung from panel J. ($n \geq 4$ per group). Scale bar: 5 μm. (Q) Quantification of panel P. The percentage of cells with γH2AX foci was calculated. (R) Quantification of panel P. The average number of γH2AX foci per cell. Data information: All values are mean ± SEM. The unpaired Student's two-tailed *t*-test was used to determine the statistical significance (*$P < 0.05$, **$P < 0.01$, ***$P < 0.001$, ****$P < 0.0001$). $n =$ number of biological replicates. Source data are available online for this figure.

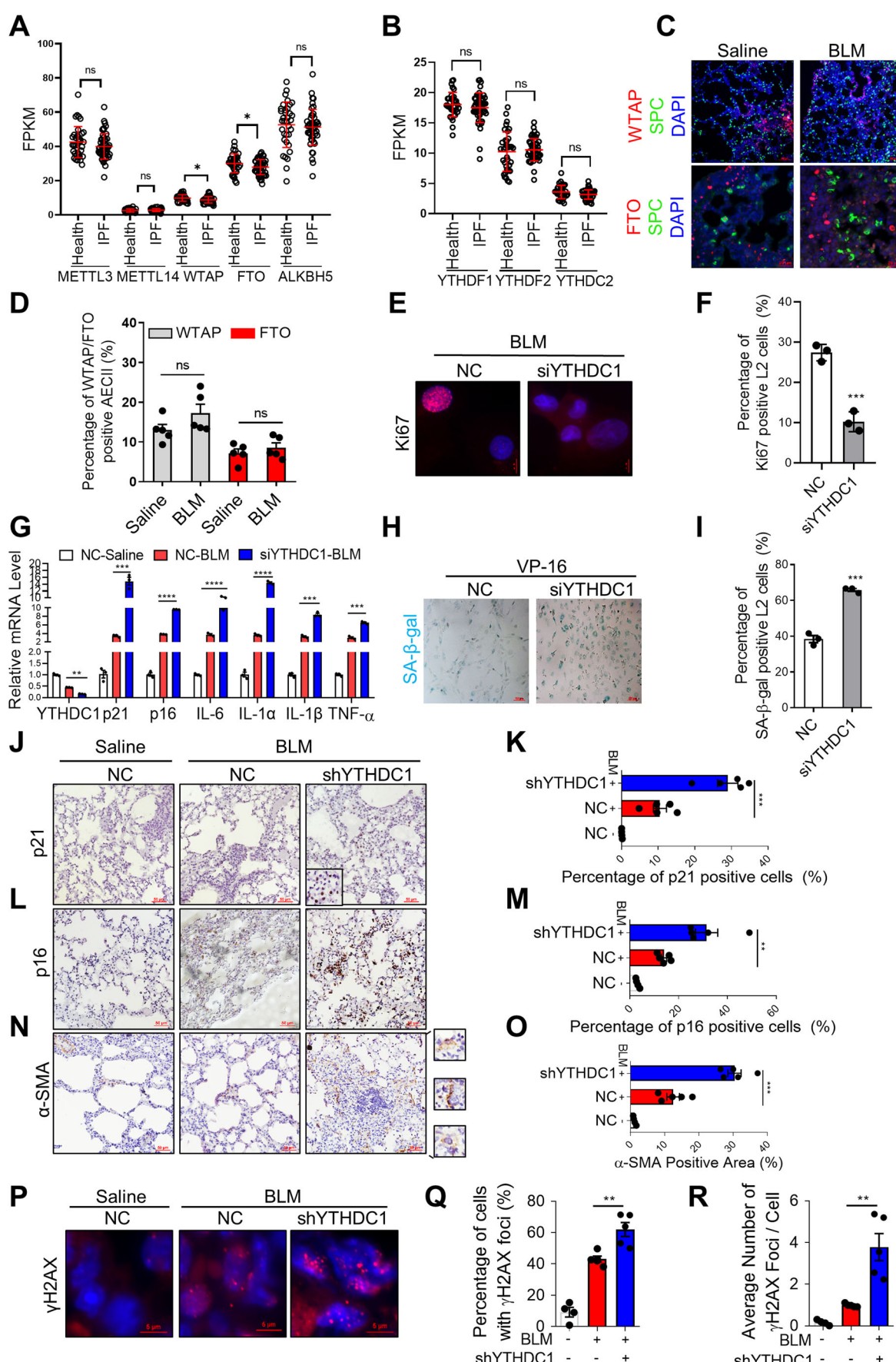

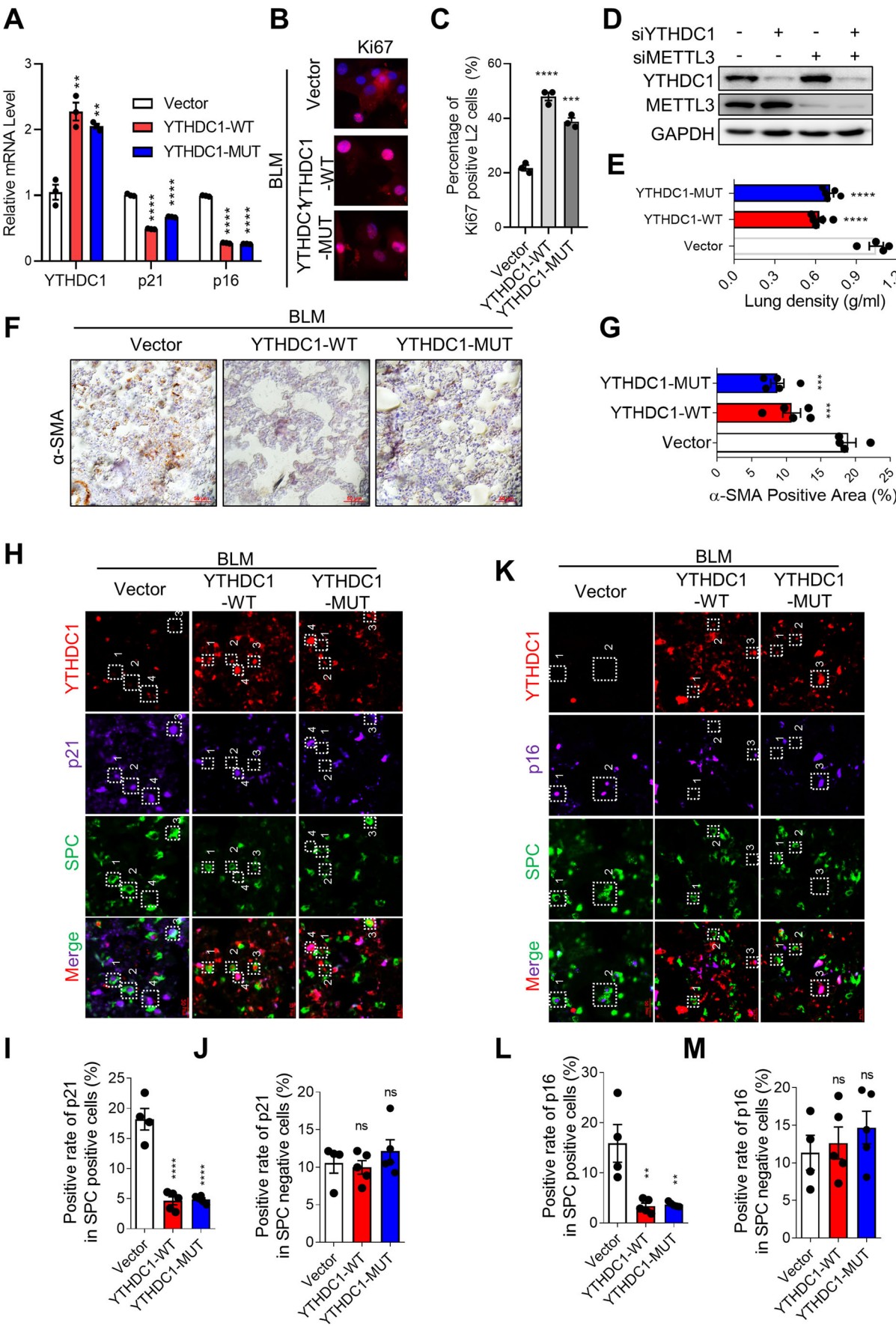

◄

**Figure EV2. YTHDC1 counteracts stress-induced pulmonary epithelial cell senescence independent of its m6A binding activity.**

(A) RT-qPCR analysis of YTHDC1, p21 and p16 mRNA level in the L2 cells overexpressed with Vector, YTHDC1-WT or YTHDC1-MUT. Forty-eight hours after transfection, cells were treated with BLM for 4 days and subjected to RT-qPCR. ($n = 3$). (B) IF detection of Ki67 in L2 cells overexpressed with Vector, YTHDC1-WT or YTHDC1-MUT. Cells were treated with BLM for 4 days. Scale bar: 10 µm. (C) Quantification of **B**. The percentage of Ki67 positive cells was calculated ($n \geq 100$ cells × three repeats). (D) Immunoblot analysis of YTHDC1 and METTL3 in L2 cells transfected with indicated siRNAs. Forty-eight hours after transfection, cells were treated with BLM for 4 days prior to analysis. ($n = 3$). (E) Quantification of mice lung density. C57/BL6 mice transfected with indicated AAV vectors were treated with BLM for 7 days ($n \geq 4$ per group). (F) Representative images of α-SMA IHC in the mice lungs from panel **E**. ($n \geq 4$ per group). Scale bar: 50 µm. (G) Quantification of panel **F**. The percentage of α-SMA positive cells was calculated. (H) Immunofluorescence was performed to determine the localization of YTHDC1 (red), p21 (purple) and SPC (green) in mice lungs from panel **E**. ($n \geq 4$ per group). Scale bar: 20 µm. (I) Quantification of panel **H**. The percentage of p21 positive cells in SPC positive cells was calculated ($n \geq 4$ per group). (J) Quantification of panel **H**. The percentage of p21 positive cells in SPC negative cells was calculated ($n \geq 4$ per group). (K–M) as in panels **H–J** except using p16 antibody to perform the IF. ($n \geq 4$ per group). Scale bar: 20 µm. Data information: All values are mean ± SEM. The One-way ANOVA was used to determine the statistical significance (**$P < 0.01$, ***$P < 0.001$, ****$P < 0.0001$). $n =$ number of biological replicates except. Source data are available online for this figure.

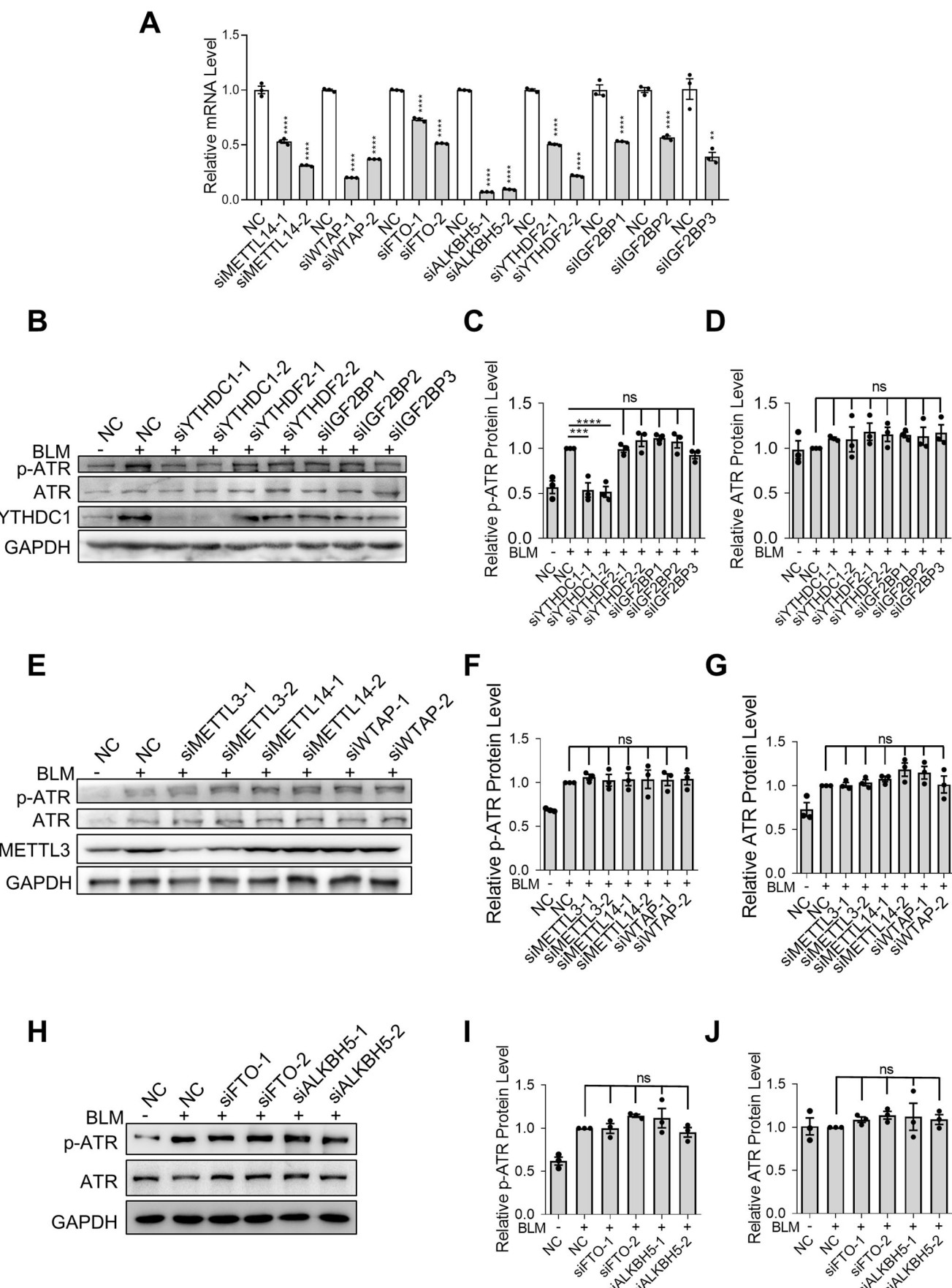

**Figure EV3.  YTHDC1 regulates the activation of ATR.**

(**A**) RT-qPCR analysis of METTL14, WTAP, FTO, ALKBH5, YTHDF2, IGF2BP1, IGF2BP2 and IGF2BP3 mRNA level in A549 cells transfected with indicated siRNAs. ($n = 3$). (**B**) Immunoblot analysis of activated ATR, total ATR and YTHDC1 in A549 cells transfected with indicated siRNAs. A549 cells were treated with BLM or saline for 4 h prior to analysis. (**C,D**), Quantification of panel **B**. The relative protein p-ATR or ATR levels were determined by normalizing the intensities of p-ATR or ATR to the intensity of GAPDH. ($n = 3$). (**E**) Immunoblot analysis of activated ATR, total ATR and METTL3 in A549 cells transfected with indicated siRNAs. A549 cells were treated with BLM or saline for 4 h prior to analysis. (**F,G**) Quantification of panel **E**. The relative protein p-ATR or ATR levels were determined by normalizing the intensities of p-ATR or ATR to the intensity of GAPDH. ($n = 3$). (**H**) Immunoblot analysis of activated ATR and total ATR in A549 cells transfected with indicated siRNAs. A549 cells were treated with BLM or saline for 4 h prior to analysis. (**I,J**), Quantification of panel **H**. The relative protein p-ATR or ATR levels were determined by normalizing the intensity of p-ATR or ATR to the intensities of GAPDH. ($n = 3$). Data information: All values are mean ± SEM. The One-way ANOVA was used to determine the statistical significance (**$P < 0.01$, ***$P < 0.001$, ****$P < 0.0001$). $n =$ number of biological replicates. Source data are available online for this figure.

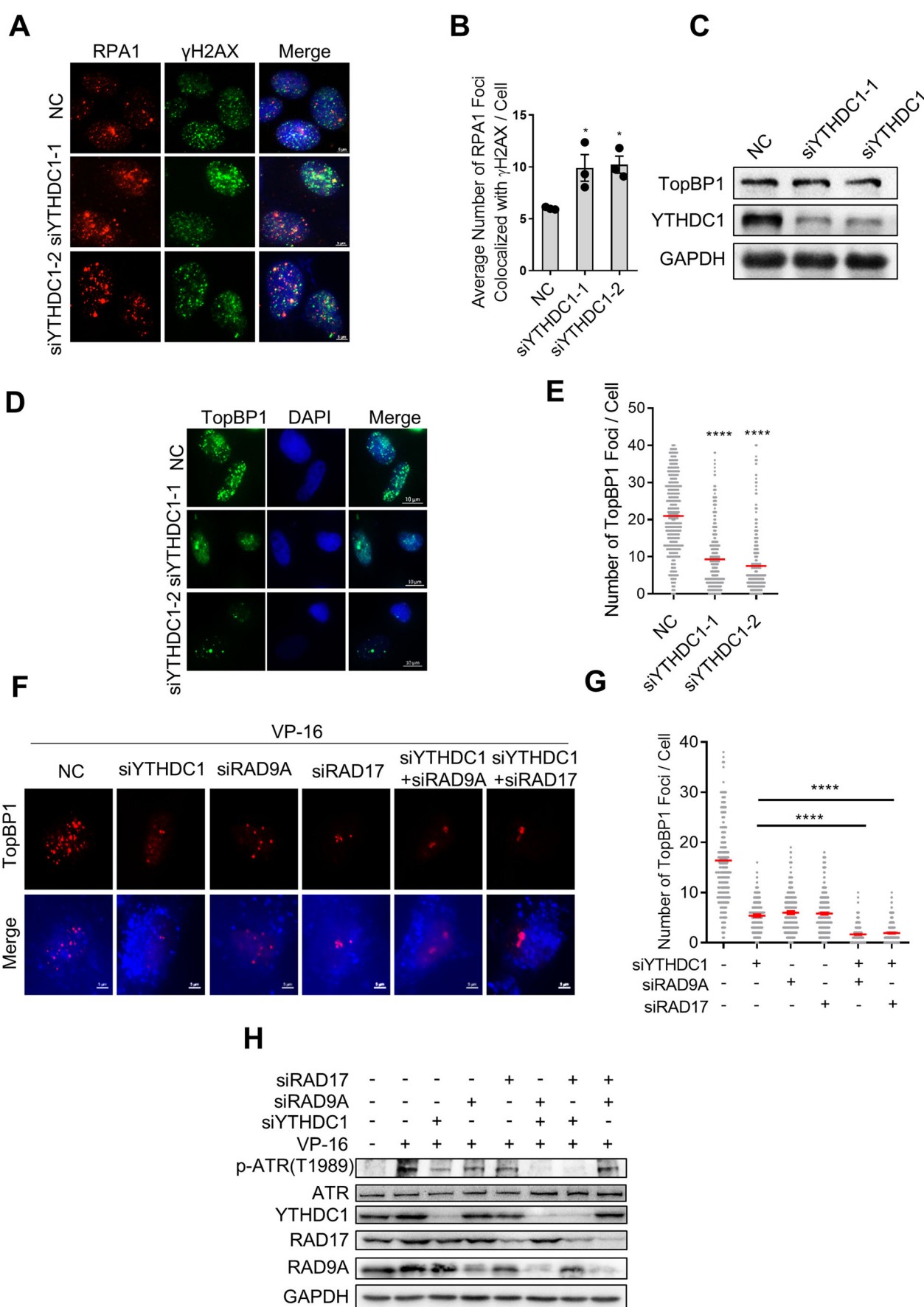

◀ **Figure EV4.  YTHDC1 regulates the recruitment of TopBP1 to activate ATR independent of 9-1-1 complex.**

(A) IF detection of RPA1 and γH2AX foci in control or YTHDC1-depleted A549 cells. Cells were treated with VP-16 for 24 h. Scale bar: 5 µm. (B) Quantification of **A**. The average number of RPA1 foci colocalized with γH2AX foci per cell ($n \geq 100$ cells × three repeats). (C) Immunoblot analysis of TopBP1 and YTHDC1 in A549 cells transfected with NC, siYTHDC1-1 or siYTHDC1-2. A549 cells were treated with VP-16 for 24 h prior to analysis ($n = 3$). (D) Immunofluorescence (IF) detection of TopBP1 foci in control or YTHDC1-depleted A549 cells. Cells were treated with BLM for 4 h before detection. Scale bar: 5 µm. (E) Quantification of **D**. The number of TopBP1 foci per cell ($n \geq 100$ cells × three repeats). (F) IF detection of TopBP1 foci in A549 cells transfected with indicated siRNAs. Cells were treated with VP-16 for 24 h. Scale bar: 5 µm. (G) Quantification of **F**. The number of TopBP1 foci per cell ($n \geq 100$ cells × three repeats). (H) Immunoblot analysis of activated ATR, total ATR, YTHDC1, RAD17, and RAD9A in A549 cells transfected with indicated siRNAs. Cells were treated with VP-16 or DMSO for 24 h prior to analysis. ($n = 3$). Data information: All values are mean ± SEM. The One-way ANOVA was used to determine the statistical significance (*$P < 0.05$, **$P < 0.01$, ****$P < 0.0001$). $n =$ number of biological replicates. Source data are available online for this figure.

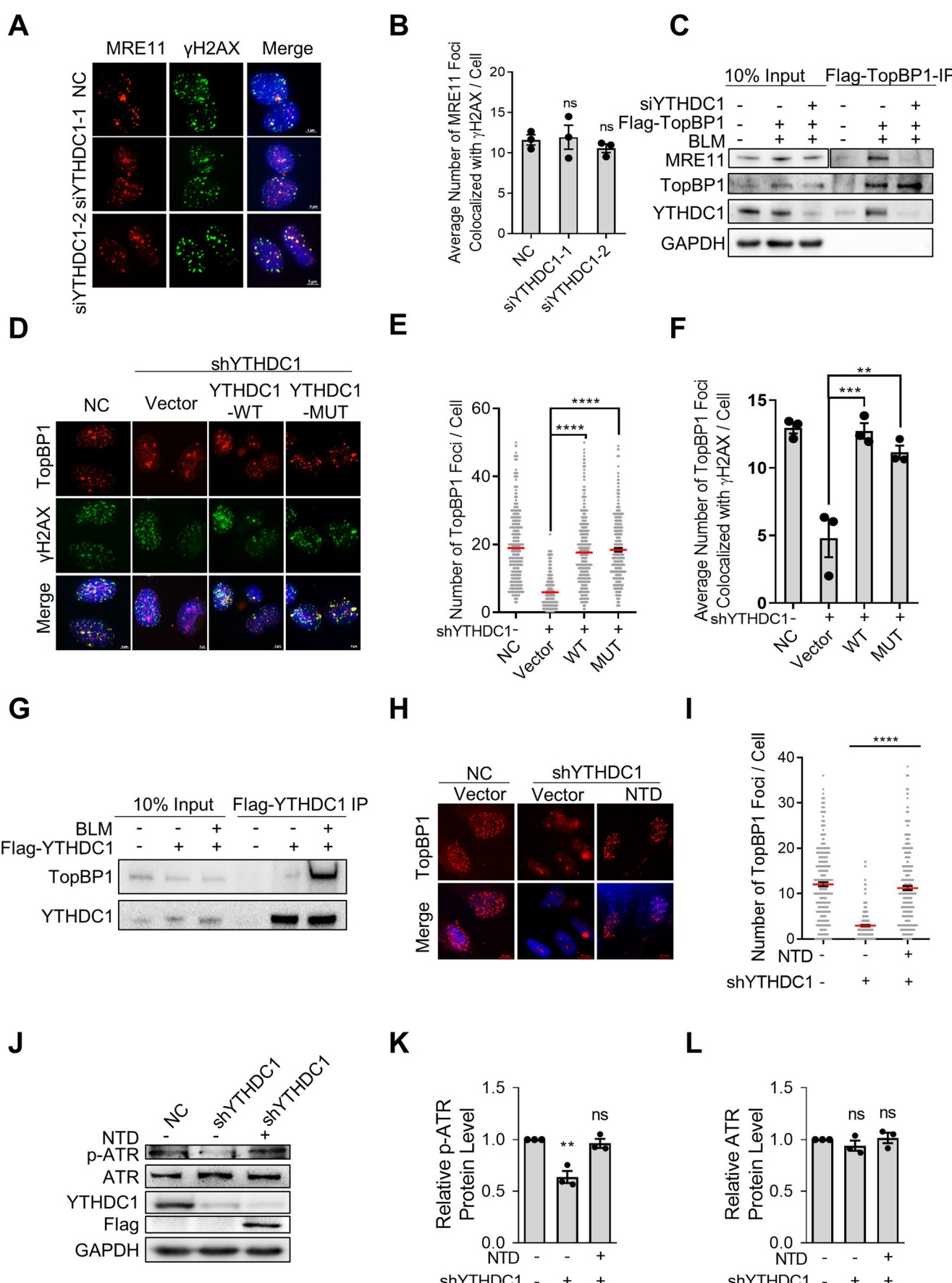

◀

**Figure EV5.  YTHDC1 regulates the recruitment of TopBP1 independent of its m6A binding activity.**

(A) IF detection of MRE11 and γH2AX foci in control or YTHDC1-depleted A549 cells. Cells were treated with VP-16 for 24 h. Scale bar: 5 μm. (B) Quantification of **A**. The average number of MRE11 foci colocalized with γH2AX foci per cell ($n \geq 100$ cells × three repeats). (C) Co-IP assay to determine the interaction of TopBP1 with MRE11 and YTHDC1 in HEK293T cells. YTHDC1 depleted cells were transfected with Flag-TopBP1 or Vector and treated with BLM for 4 h. Cell lysates were used for immunoprecipitation with Flag-beads. Immunoprecipitates were immunoblotted with MRE11, TopBP1, YTHDC1 and GAPDH antibody, respectively. ($n = 3$). (D) IF detection of TopBP1 and γH2AX foci in YTHDC1-depleted A549 cells overexpressed with Vector, YTHDC1-WT or YTHDC1-MUT. Cells were treated with VP-16 for 24 h. Scale bar: 5 μm. (E) Quantification of **D**. The number of TopBP1 foci per cell ($n \geq 100$ cells × three repeats). (F), Quantification of **D**. The average number of TopBP1 foci colocalized with γH2AX foci per cell ($n \geq 100$ cells × three repeats). (G) Co-IP assay to determine the interaction between YTHDC1 and TopBP1. HEK293T cells transfected with Flag-YTHDC1 were treated with BLM for 4 h. Cell lysates were used for immunoprecipitation with Flag-beads. Immunoprecipitates were immunoblotted with TopBP1 and YTHDC1 antibody, respectively. ($n = 3$). (H) IF detection of TopBP1 foci in YTHDC1 defect A549 cells overexpressed Vector or NTD of YTHDC1. Forty-eight hours after transfection, cells were treated with VP-16 for 24 h. Scale bar: 5 μm. (I) Quantification of **H**. The number of TopBP1 foci per cell ($n \geq 100$ cells × three repeats). (J) Immunoblot analysis of activated ATR, total ATR, YTHDC1 and Flag from panel **H**. (K,L) Quantification of panel **J**. The relative protein p-ATR or ATR levels were determined by normalizing the intensities of p-ATR or ATR to the intensity of GAPDH. ($n = 3$). Data information: All values are mean ± SEM. The unpaired Student's two-tailed *t*-test was used to determine the statistical significance between two groups. The One-way ANOVA was used to determine the statistical significance for more than two groups (**$P < 0.01$, ***$P < 0.001$, ****$P < 0.0001$). $n =$ number of biological replicates. Source data are available online for this figure.

