## [Peer Review File · The EMBO Journal]

YTHDC1 delays cellular senescence and pulmonary fibrosis by activating ATR in an m6A-independent manner

Canfeng Zhang, Liping Chen, Chen Xie, Fengwei Wang, Juan Wang, Haoxian Zhou, Qianyi Liu, Zhuo Zeng, Na Li, Junjiu Huang, Yong Zhao, and Haiying Liu

DOI: [10.15252/emboj.2023113675](https://doi.org/10.15252/emboj.2023113675)

Corresponding author: Haiying Liu (liuhy5@mail.sysu.edu.cn)

Review Timeline:

Submission Date:	2nd Feb 23
Editorial Decision:	12th Mar 23
Revision Received:	26th May 23
Editorial Decision:	26th Jul 23
Revision Received:	23rd Sep 23
Accepted:	26th Oct 23

Editor: Daniel Klimmeck

Transaction Report:

Dear Dr Liu,

Thank you for the submission of your manuscript (EMBOJ-2022-113675) to The EMBO Journal, as well as for your patience with our response at this time of the year. Your study has been sent to two reviewers with expertise in fibrosis, senescence and DNA damage, and we have received feedback from both of them, which I enclose below.

As you will see, the referees acknowledge the potential interest and novelty of your results, although they also express several issues that will have to be conclusively addressed before they can be supportive of publication of your manuscript in The EMBO Journal. In more detail, the reviewers raise a number of points related to additional experiments and controls required, improved methods annotation and data processing, overall discussion and acknowledgement of the limitations of the findings, that would need to be conclusively addressed to achieve the level of robustness and clarity needed for The EMBO Journal.

I judge the comments of the referees to be generally reasonable and given their overall interest, we are in principle happy to invite you to revise your manuscript experimentally to address the referees' comments.

As you may remember from previous experience, we generally allow three months as standard revision time. As a matter of policy, competing manuscripts published during this period will not negatively impact on our assessment of the conceptual advance presented by your study. However, we request that you contact the editor as soon as possible upon publication of any related work, to discuss how to proceed. Should you foresee a problem in meeting this three-month deadline, please let us know in advance and we may be able to grant an extension.

When submitting your revised manuscript, please carefully review the instructions below.

Thank you for the opportunity to consider your work for publication.
I look forward to your revision.

Best regards,

Daniel Klimmeck

Daniel Klimmeck, PhD
Senior Editor
The EMBO Journal

Instruction for the preparation of your revised manuscript:

- 1) a .docx formatted version of the manuscript text (including legends for main figures, EV figures and tables). Please make sure that the changes are highlighted to be clearly visible.
- 2) individual production quality figure files as .eps, .tif, .jpg (one file per figure).
- 3) a .docx formatted letter INCLUDING the reviewers' reports and your detailed point-by-point response to their comments. As part of the EMBO Press transparent editorial process, the point-by-point response is part of the Review Process File (RPF), which will be published alongside your paper.
- 4) a complete author checklist, which you can download from our author guidelines ([https://wol-prod-cdn.literatumonline.com/pb-assets/embo-site/Author Checklist%20-%20EMBO%20J-1561436015657.xlsx](https://wol-prod-cdn.literatumonline.com/pb-assets/embo-site/Author%20Checklist%20-%20EMBO%20J-1561436015657.xlsx)). Please insert information in the checklist that is also reflected in the manuscript. The completed author checklist will also be part of the RPF.
- 5) Please note that all corresponding authors are required to supply an ORCID ID for their name upon submission of a revised manuscript.
- 6) It is mandatory to include a 'Data Availability' section after the Materials and Methods. Before submitting your revision, primary datasets produced in this study need to be deposited in an appropriate public database, and the accession numbers and database listed under 'Data Availability'. Please remember to provide a reviewer password if the datasets are not yet public (see <https://www.embopress.org/page/journal/14602075/authorguide#datadeposition>). In case you have no data that requires deposition in a public database, please state so in this section. Note that the Data Availability Section is restricted to new primary data that are part of this study.
*** Note - All links should resolve to a page where the data can be accessed. ***

7) Our journal encourages inclusion of *data citations in the reference list* to directly cite datasets that were re-used and obtained from public databases. Data citations in the article text are distinct from normal bibliographical citations and should directly link to the database records from which the data can be accessed. In the main text, data citations are formatted as follows: "Data ref: Smith et al, 2001" or "Data ref: NCBI Sequence Read Archive PRJNA342805, 2017". In the Reference list, data citations must be labeled with "[DATASET]". A data reference must provide the database name, accession number/identifiers and a resolvable link to the landing page from which the data can be accessed at the end of the reference. Further instructions are available at .

8) We would also encourage you to include the source data for figure panels that show essential data. Numerical data can be provided as individual .xls or .csv files (including a tab describing the data). For 'blots' or microscopy, uncropped images should be submitted (using a zip archive or a single pdf per main figure if multiple images need to be supplied for one panel). Additional information on source data and instruction on how to label the files are available at .

9) We replaced Supplementary Information with Expanded View (EV) Figures and Tables that are collapsible/expandable online (see examples in <https://www.embopress.org/doi/10.15252/embj.201695874>). A maximum of 5 EV Figures can be typeset. EV Figures should be cited as 'Figure EV1, Figure EV2' etc. in the text and their respective legends should be included in the main text after the legends of regular figures.

10) When assembling figures, please refer to our figure preparation guideline in order to ensure proper formatting and readability in print as well as on screen:
<http://bit.ly/EMBOPressFigurePreparationGuideline>

11) For data quantification: please specify the name of the statistical test used to generate error bars and P values, the number (n) of independent experiments (specify technical or biological replicates) underlying each data point and the test used to calculate p-values in each figure legend. The figure legends should contain a basic description of n, P and the test applied. Graphs must include a description of the bars and the error bars (s.d., s.e.m.).

Further information is available in our Guide to Authors: <https://www.embopress.org/page/journal/14602075/authorguide>

We realize that it is difficult to revise to a specific deadline. In the interest of protecting the conceptual advance provided by the work, we recommend a revision within 3 months (10th Jun 2023). Please discuss the revision progress ahead of this time with the editor if you require more time to complete the revisions.

Referee #1:

In this manuscript authors propose that YTHDC1 counteracts pulmonary senescence and disease. They show that YTHDC1 primarily expresses in pulmonary alveolar epithelial type 2 (AECII) cells and when mice are treated with bleomycin (a model of IPF), YTHDC1 expression is significantly decreased in AECII. Authors then suggest that YTHDC1 depletion accelerates senescence and fibrosis in the lung after BLM and its overexpression alleviates pulmonary senescence and fibrosis, Finally, authors conduct mechanistic studies which propose that YTHDC1 promotes the interaction between TopBP1 and MRE11, thus activating the ATR and facilitating repair of DNA damage.

I think the manuscript describes an interesting and potentially important finding. However, I believe there is much room for improvement. There are limitations inherent to the bleomycin model and while authors show reduced YTHDC1 at mRNA level in IPF patients- I believe a more detailed characterization of IPF patients should be conducted. Additionally, the evidence for senescence can be strengthened by the use of additional markers- particularly in the in vivo studies. Additionally, the only lung phenotype analyzed was fibrosis using Masson's trichrome staining. Finally, why AECII cells are examined rather than other cell-types needs some more justification- bleomycin will cause senescence in many other cell-types in the lung. Why are AECII cells the only important ones in inducing fibrosis?

Overall, I believe the manuscript would also benefit from some rewriting and more clear explanation for the rationale of the experiments. Language use at times could be improved for clarity.

Authors claim that "accumulated senescent AECII cells is one of the driving factors of IPF" and cite a manuscript by Yao et al. 2021 that shows conditional loss of Sin3a in adult mouse AT2 cells initiates a program of p53-dependent cellular senescence and leads to fibrosis. Other cell-types in IPF have also been shown to show senescent markers. Also- surprised author did not refer to manuscript Schafer et al. 2017 that demonstrated that clearance of p16 -positive senescent cells (using a transgenic mouse model) improved phenotypes in bleomycin induced lung fibrosis.

Characterization of senescence- authors measured Sa-b-gal, loss of Ki67, p21 and some SASP components in cells. What about p16?

In mice- authors performed SA-b-Gal and gH2A.X as markers of senescence in mice treated with BLM after KD of YTHDC1 . Not clear how they can establish from the images presented that AECii cells stain positive for the markers. What about other cell types? Other senescent markers should be examined. SA-b-Gal can be present in conditions not associated with senescence as several studies have previously demonstrated. Authors should measure either by staining or qPCR in tissues levels of p21, p16 and SASP factors. I find the gH2A.X staining to be uncharacteristic- I would expect the presence of distinct foci in nuclei, not a homogeneous nuclear staining. Authors measure fibrosis using Masson's trichrome staining- could other complementary methods to examine fibrosis be done? Also is the SASP induced in these senescent cells pro-fibrotic? Any other measurements of lung function would also be desirable.

Similar issues are raised with regards to Figure 2. Also, why was IL-1a examined in Figure 1 but not Figure 2? Authors should be consistent and not only show the data that fits their hypothesis.

Why were different cell-types used in different experiments. Also, why in some experiments etoposide was used to induced DNA damage and not bleomycin?

The role of YTHDC1 in ATR activation. Why the 24hour time point was chosen- why not a kinetics of repair was shown? None of the western blots were quantified (how many independent experiments were performed?)

p-ATR staining is not particularly convincing in mice lungs (distinct foci in nuclei).

Some better examples (magnifications) of co-localization between TopBP1 and gH2A.X should be shown.

Statistics need very careful revision- authors used unpaired Student's two-tailed t-test for multiple comparisons which is not correct.

Referee #2:

This manuscript describes the m6A binding-independent role of YTHDC1 in maintaining genomic stability and protecting lung against stress-induced senescence and fibrosis.

In mechanism, YTHDC1 can directly binds to TopBP1 to regulate TopBP1 interacting with MRE11 and then activates DNA damage repair. The authors also demonstrated reduced YTHDC1 expression in both IPF patients and bleomycin induced pulmonary fibrosis mice model, and overexpress YTHDC1 can prevent fibrosis in pulmonary fibrosis mice. These results are of potential interest since they suggest that YTHDC1 is an important regulator in senescence and provides a potential therapeutic target for treating pulmonary fibrosis. However, much of the analysis is based on immunostaining, and western blots data lack quantification. The authors need to provide further evidence for their conclusions.

Specific Comments:

1. Publicly available and interactive lung single cell dataset such as LungMap show that ythdc1 express widely in all lung cell types both for human and mouse, would the authors care to speculate the reason that YTHDC1 primarily expresses in AECII cells in normal lung tissues.
2. The authors harvested the lung 7 days after bleomycin to assess lung fibrosis, however, the most suitable time point for assessing lung fibrosis in this model is 14 days after IT instillation of bleomycin, as 7 days the mice only have mild fibrosis.
3. When the authors knocked down YTHDC1 in mice lungs using AAV6 expressing system, is there any spontaneous fibrosis phenotype?
4. The authors only use SA-b-gal staining to quantify senescence, additional senescent cell markers like P21 staining should also be performed.
5. The western blot data need quantification to further support the conclusion.
6. Figure 1a showed that YTHDC1 significantly decreases during lung fibrosis in bleo treated mice, however, in figure S3b, YTHDC1 significantly increase in bleo treated cells, is this the time difference? The authors should measure the YTHDC1 level at different time points.

7. It is hard to visualize in the images provided in Figure S1C, as the authors stated that WTAP and FTO universally express in different kinds of cells besides AECII.

Response to the comments of Reviewer #1

Reviewer #1: In this manuscript authors propose that YTHDC1 counteracts pulmonary senescence and disease. They show that YTHDC1 primarily expresses in pulmonary alveolar epithelial type 2 (AECII) cells and when mice are treated with bleomycin (a model of IPF), YTHDC1 expression is significantly decreased in AECII. Authors then suggest that YTHDC1 depletion accelerates senescence and fibrosis in the lung after BLM and its overexpression alleviates pulmonary senescence and fibrosis, Finally, authors conduct mechanistic studies which propose that YTHDC1 promotes the interaction between TopBP1 and MRE11, thus activating the ATR and facilitating repair of DNA damage. I think the manuscript describes an interesting and potentially important finding. However, I believe there is much room for improvement.

We thank the reviewer for his/her critical reading of the manuscript and for his/her thoughtful suggestions. Please see our responses to the specific issues below.

Comment 1: There are limitations inherent to the bleomycin model and while authors show reduced YTHDC1 at mRNA level in IPF patients- I believe a more detailed characterization of IPF patients should be conducted.

Response: IPF is a chronic, progressive lung disease with accumulated senescent cells and becomes damaged and scarred. Although different animal models have been constructed to investigate the IPF pathology, no one can recapitulate all features of this complicated disease. Here we used the currently most extensively used bleomycin model which displays accumulated DNA damages and senescent cells at the early stage, and fit for studying the role of stress induced senescence in IPF development. It is attractive to study the YTHDC1 protein expression levels and its impact on IPF patients. We are regret that currently we are not able to conduct it for patients are not available. For the dataset used in the study, the detailed information of IPF patients is carefully described in revised method (Page 26).

Comment 2: Additionally, the evidence for senescence can be strengthened by the use of additional markers- particularly in the in vivo studies.

Response: We thank the reviewer for this suggestion. P21 and p16 have been detected by IHC experiment in mice lungs transfected with shYTHDC1 AAV or

YTHDC1 overexpressed AAV (Fig EV1J-M and Fig EV2F-I). In addition, SASP factors such as IL-1 α , IL-8 and TGF- β were examined by RT-qPCR (Fig 1I and Fig 2G). These results support the idea that YTHDC1 plays a protective role during pulmonary senescence. These results are included in revised MS (Page 7-8).

Comment 3: Additionally, the only lung phenotype analyzed was fibrosis using Masson's trichrome staining.

Response: As suggested, other complementary methods have been done to strengthen the phenotype of fibrosis. We have shown that depletion of YTHDC1 leads to significant increase of lung density (Fig 1G-H) and the levels of fibrotic markers such as α -smooth muscle actin (α -SMA) (Fig EV1N-O), Vimentin and COL1A (Fig 1I). Moreover, overexpression of YTHDC1-WT and YTHDC1-MUT result in less lung density (Fig EV2E) and the level of α -SMA (Fig EV2J-K), Vimentin and COL1A (Fig 2G). These results are consistent with and strengthen the conclusion obtained from Masson's trichrome staining. We thank the reviewer for his/her suggestion. These results are included in revised MS (Page 7-8).

Comment 4: Finally, why AECII cells are examined rather than other cell-types needs some more justification- bleomycin will cause senescence in many other cell-types in the lung. Why are AECII cells the only important ones in inducing fibrosis?

Response: Several Studies have shown that the senescence of AECs (primarily AECII), fibroblasts and myofibroblasts are involved in the occurrence and development of pulmonary fibrosis (1-4). Although the BLM induces the senescence of all these cells, it has reported that AECII injury/dysfunction serves as an early initiating event in pulmonary fibrosis (PF) that leads to fibroproliferation and progressive loss of lung function (5). A previous review showed that AECII, as the main source of pro-fibrogenic cytokines in PF, express a variety of cytokines and growth factors, which can promote the migration, proliferation and accumulation of extracellular matrix of fibroblasts (4). In our study, we did not intend to emphasize that the senescence of AECII is the most important or the only factor to cause pulmonary fibrosis. Nevertheless, in the process of fibrosis, we found that the expression of YTHDC1 is mainly decreased in AECII, rather than in fibroblasts (Picture 1). Therefore, we mainly focused on the function of YTHDC1 in the senescence of AECII and pulmonary fibrosis and we didn't rule out the function of

YTHDC1 in fibroblasts and other cells. We feel sorry for the confusion caused by the unclear description of our MS. We have modified the confusing sentences to avoid misunderstanding (revised MS Page 3 and 6).

Picture1: The protein level of YTHDC1 in different VP-16 treated time

a, Immunofluorescence was performed to determine the localization of YTHDC1 (red) and Vimentin (green) in mice lungs that treated with saline or BLM for 7 days. Scale bar: 100 μ m. **b**, Quantification of panel **a**. The percentage of double-positive cells in YTHDC1-positive cells was calculated (n=5 per group).

Comment 5: Overall, I believe the manuscript would also benefit from some rewriting and more clear explanation for the rationale of the experiments. Language use at times could be improved for clarity.

Response: We thank the reviewer for this suggestion. We have carefully repolished the manuscript and explained more clearly on the rationale of the experiments.

Comment 6: Authors claim that "accumulated senescent AECII cells is one of the driving factors of IPF" and cite a manuscript by Yao et al. 2021 that shows conditional loss of Sin3a in adult mouse AT2 cells initiates a program of p53-dependent cellular senescence and leads to fibrosis. Other cell-types in IPF have also been shown to show senescent markers. Also- surprised author did not refer to manuscript Schafer et al. 2017 that demonstrated that clearance of p16 -positive senescent cells (using a transgenic mouse model) improved phenotypes in bleomycin induced lung fibrosis.

Response: We apologize for this confusing description. We have revised our manuscript to avoid misleading, for the details pleased refer to the response to Comment 4. The work of Schafer et al. 2017 has been briefly introduced in our manuscript (revised MS Page 3 and 6).

Comment 7: Characterization of senescence- authors measured Sa-b-gal, loss of Ki67, p21 and some SASP components in cells. What about p16?

Response: We detected the p16 expression levels as suggested. The results showed that knockdown of YTHDC1 increase the level of p16 in L2 cells (Fig EV1G). In addition, overexpression of YTHDC1-WT and YTHDC1-MUT reduce the p16 level induced by DNA damage in L2 cells (Fig EV2A). These results are included in revised MS (Page 6-7).

Comment 8: In mice- authors performed SA-b-Gal and γ H2A.X as markers of senescence in mice treated with BLM after KD of YTHDC1. Not clear how they can establish from the images presented that AECii cells stain positive for the markers. What about other cell types?

Response: The AECII marker SPC was co-stained with SA- β -gal, and the percentages of double positive cells were quantified so as to determine the senescence of AECII cells. The DNA damage marker γ H2A.X was not co-stained with SPC, and we can't tell whether the DNA damages are in AECII. The statement has been revised to avoid the confusion.

Comment 9: Other senescent markers should be examined. SA-b-Gal can be present in conditions not associated with senescence as several studies have previously demonstrated. Authors should measure either by staining or qPCR in tissues levels of p21, p16 and SASP factors.

Response: We thank the reviewer for this suggestion. P21 and p16 have been detected by IHC experiment in mice lungs transfected with shYTHDC1 AAV or YTHDC1 overexpressed AAV (Fig EV1J-M and Fig EV2F-I). In addition, SASP factors such as IL-1 α , IL-8 and TGF- β were examined by RT-qPCR (Fig 1I and Fig 2G). These results support the idea that YTHDC1 plays a protective role during pulmonary senescence. These results are included in revised MS (Page 7-8).

Comment 10: I find the γ H2A.X staining to be uncharacteristic-I would expect the presence of distinct foci in nuclei, not a homogeneous nuclear staining.

Response: Because IHC staining is visualized by chromogenic dyes with enzymatic reactions (e.g., DAB and HRP) which hardly display as distinct foci in nuclei, we

performed IF to examine the signal of γ H2A.X in mice lung tissues. The results show that the γ H2A.X forms distinct foci in nuclei after BLM treatment, and depletion of YTHDC1 increases both the average number of γ H2A.X foci and the positive rate of cells with γ H2A.X foci (Fig EV1P-R). These results have been added to the revised MS (Page 7).

Comment 11: Authors measure fibrosis using Masson's trichrome staining- could other complementary methods to examine fibrosis be done?

Response: As suggested, other complementary methods have been done to strengthen the phenotype of fibrosis. We have shown that depletion of YTHDC1 leads to significant increase of lung density (Fig 1G-H) and the levels of fibrotic markers such as α -smooth muscle actin (α -SMA) (Fig EV1N-O), Vimentin and COL1A (Fig 1I). Moreover, overexpression of YTHDC1-WT and YTHDC1-MUT result in less lung density (Fig EV2E) and the level of α -SMA (Fig EV2J-K), Vimentin and COL1A (Fig 2G). These results are consistent with and strengthen the conclusion obtained from Masson's trichrome staining. We thank the reviewer for his/her suggestion. These results are included in revised MS (Page 7-8).

Comment 12: Also is the SASP induced in these senescent cells pro-fibrotic?

Response: Yes, lots of studies have reported that the SASP factors expressed by senescent cells directly or indirectly promote lung fibrosis. TGF- β is probably the best-studied cytokine in fibrosis, and is regarded as a prototypical "pro-fibrotic" mediator. The TGF- β has been found that can promote fibroblast proliferation and differentiation into myofibroblasts and drive ECM accumulation, especially collagen and fibronectin, which irreversibly remodel the lung tissue structure and promote fibrosis(6,7). The evidence that the SASP pro-fibrotic also can be support by overexpression of IL-1 β in rat lungs promoted lung fibrosis characterized by the presence of myofibroblasts, fibroblast foci, and ECM accumulation. Notably, reduce expression of SASP factors through removal of senescent cells or by using monoclonal antibodies or small molecules was found to alleviate pulmonary fibrosis (8-10). Recently, it has been further demonstrated that tralokinumab, a human IL-13 neutralizing monoclonal antibody, dampened pulmonary fibrosis and promoted lung repair in a humanized severe combined immunodeficiency IPF model (11).

Comment 13: Any other measurements of lung function would also be desirable.

Response: It is attractive to measure the effect of YTHDC1 on pulmonary functions, like lung density, resting volume, capacity and minute ventilation by spirometry test using mouse lung function measuring instruments such as high-resolution CT and eSpira Maneuvers System. Unfortunately, we cannot get these machines to measure the lung functions. To try to address this question, we performed alternative experiment as following. We measured the lung density using buoyancy test. In general, healthy lung filling with air can float on the surface of the water, whereas in pulmonary fibrosis, some areas that should be filled with air become substantial, thus making the lung tissue heavier and denser, and therefore sink to the bottom of the water. Through this method, we found that YTHDC1 deletion increased lung tissue density in BLM-treated mice (Fig 1G-H), while overexpression of YTHDC1-WT and YTHDC1-MUT reduced BLM-induced lung density increase (Fig EV2E). These results have been added to the revised MS (Page 7-8).

Comment 14: Similar issues are raised with regards to Fig 2. Also, why was IL-1a examined in Fig 1 but not Fig 2? Authors should be consistent and not only show the data that fits their hypothesis.

Response: As described above, in the revised MS most experiments done in Fig 1 have also been done in Fig 2. The IL-1 α was examined by RT-qPCR in Fig 2G. Thanks for pointing it out.

Comment 15: Why were different cell-types used in different experiments.

Response: We used L2 to investigate the function of YTHDC1 in senescence because it is a normal rat type II alveolar epithelial cell and has limited proliferative capacity. The A549 and HEK293T are cancer cells or immortalized cells that can proliferate indefinitely and are easy to transfection. We used the A549 and HEK293T to study the mechanism by which YTHDC1 regulates cellular senescence.

Comment 16: Also, why in some experiments etoposide was used to induced DNA damage and not bleomycin?

Response: Both VP-16 (etoposide) and BLM are widely used DNA damage inducers. In our study, we used both VP-16 and BLM to detect the function of YTHDC1 in stress induced senescence and ATR activation. In the original MS, we only used VP-

16 treatment for mechanism study, and in the revised MS we repeated the key experiments using BLM treatment, including the number of Topbp1 foci (Fig EV4D-E), the interaction between TopBP1 and MRE11 (Fig EV5C) and the interaction between TopBP1 and YTHDC1 (Fig EV6A). All of these experiments get similar results with VP-16 treatment. These results are included in revised MS (Page 10-12).

Comment 17: The role of YTHDC1 in ATR activation. Why the 24 hour time point was chosen- why not a kinetics of repair was shown?

Response: We have detected the ATR activation at different time points. The results showed that 24 h treatment displayed highest level of ATR activation (Picture 2). In addition, 24-hour treatment showed the highest YTHDC1 levels, therefore resulting in most significant knockdown efficiency of YTHDC1 (Picture 2).

Picture2: The protein level of YTHDC1 in different VP-16 treated time

a, Immunoblot analysis of YTHDC1 in A549 cells. A549 cells depletion of YTHDC1 were treated with VP-16 for indicated times prior to analysis. Quantitative values are the mean ± SEM of n = 3 experiments.

Comment 18: None of the western blots were quantified (how many independent experiments were performed?)

Response: The western blots of Fig 3A, D, G, J M, Fig 4I, Fig EV3B, E, H and Fig EV6D have been quantified. All of the western-blot experiments have been repeated three times. We thank the reviewer for bringing this to our attention.

Comment 19: p-ATR staining is not particularly convincing in mice lungs (distinct foci in nuclei).

Response: As suggested, we performed IF to stain the p-ATR in mice lungs treated with BLM. The results showed that about 40% of the cells could form punctate

nuclear foci after exposure to BLM and each cell contains 3 p-ATR foci on average. Knockdown of YTHDC1 significantly impaired the positive rate and foci number of p-ATR (Fig 3P-R and revised MS page 9).

Comment 20: Some better examples (magnifications) of co-localization between TopBP1 and γ H2A.X should be shown.

Response: Thanks. The Fig 4e has been revised to show co-localization between TopBP1 and γ H2A.X clearer.

Comment 21: Statistics need very careful revision- authors used unpaired Student's two-tailed t-test for multiple comparisons which is not correct.

Response: It has been corrected. In Fig 2, Fig 3, Fig 4F-G, Fig EV2, Fig EV3, Fig EV4B, E, Fig EV5E-F and Fig EV6E-F, one-way ANOVA has been used for statistic comparison. We thank the reviewer for bringing this to our attention.

Response to the comments of Reviewer #2

This manuscript describes the m6A binding-independent role of YTHDC1 in maintaining genomic stability and protecting lung against stress-induced senescence and fibrosis.

In mechanism, YTHDC1 can directly binds to TopBP1 to regulate TopBP1 interacting with MRE11 and then activates DNA damage repair. The authors also demonstrated reduced YTHDC1 expression in both IPF patients and bleomycin induced pulmonary fibrosis mice model, and overexpress YTHDC1 can prevent fibrosis in pulmonary fibrosis mice. These results are of potential interest since they suggest that YTHDC1 is an important regulator in senescence and provides a potential therapeutic target for treating pulmonary fibrosis. However, much of the analysis is based on immunostaining, and western blots data lack quantification. The authors need to provide further evidence for their conclusions.

Comment 1: Publicly available and interactive lung single cell dataset such as LungMap show that *ythdc1* express widely in all lung cell types both for human and mouse, would the authors care to speculate the reason that YTHDC1 primarily expresses in AECII cells in normal lung tissues.

Response: As suggested, we have looked over the mRNA expression level of YTHDC1 in different mice lung cells using data from LungMap, and found there is

not much difference in mRNA level. It is probably attributed to two reasons. Firstly, sometimes the protein levels are different whereas the mRNA levels are the same due to translational regulation. We use IF to measure the YTHDC1 protein level, and probably AECII only expresses more YTHDC1 at protein level. Secondly, the data from LungMap came from mice at the embryonic stage or a few days after birth, whereas our experiments were performed with 8-week-old mice. The expression of YTHDC1 probably changes with age increasing.

Comment 2: The authors harvested the lung 7 days after bleomycin to assess lung fibrosis, however, the most suitable time point for assessing lung fibrosis in this model is 14 days after IT instillation of bleomycin, as 7 days the mice only have mild fibrosis.

Response: We thank the reviewer for bringing this to our attention. The principle for bleomycin induction of pulmonary fibrosis is that it causes DNA damage and inflammation in cells, leading to apoptosis of AECI, and the supplement of AECI depends on the proliferation and differentiation of AECII. However, K. Aoshiba et al. showed that senescent AEC II caused by bleomycin treatment reaches a maximum at day 7 (12). In our study, we mainly focused on the role of YTHDC1 in the senescence process of AECII and found that depletion of YTHDC1 enhanced senescence of AECII. Therefore, samples were collected 7 days after bleomycin injection. The reason and reference have been included in our revised MS (Page 7).

Comment 3: When the authors knocked down YTHDC1 in mice lungs using AAV6 expressing system, is there any spontaneous fibrosis phenotype?

Response: To determine whether knockdown of YTHDC1 by AAV6 causes spontaneous lung fibrosis, we performed RT-qPCR to examine the expression of fibrosis related-gene Vimentin and Colla1, and IHC to detect the level of α -SMA, a marker of lung fibrosis. The results showed that depletion of YTHDC1 using AAV did not increase the level of Vimentin, Colla1 or α -SMA (picture3 a-c), which indicated that depletion of YTHDC1 did not cause any spontaneous fibrosis phenotype.

Picture3: Depletion of YTHDC1 does not cause any spontaneous fibrosis phenotype

a, RT-qPCR analysis of YTHDC1, Vimentin and Col1a1 mRNA level in mice lungs. C57/BL6 mice transfected with indicated shAAV vectors were treated with saline for 7 days (n≥4 per group). **b**, Representative images of α -SMA IHC in the mice lungs from panel a. (n≥4 per group). Scale bar: 50 μ m. **c**, Quantification of panel b. The percentage of α -SMA positive cells was calculated. The unpaired Student's two-tailed t-test was used to determine the statistical significance (*P<0.05).

Comment 4: The authors only use SA-b-gal staining to quantify senescence, additional senescent cell markers like P21 staining should also be performed.

Response: We thank the reviewer for this suggestion. P21 and p16 have been detected by IHC experiment in mice lungs transfected with shYTHDC1 AAV or YTHDC1 overexpressed AAV (Fig EV1J-M and Fig EV2F-I). In addition, SASP factors such as IL-1 α , IL-8 and TGF- β were examined by RT-qPCR (Fig 1I and Fig 2G). These results support the idea that YTHDC1 plays a protective role during pulmonary senescence. These results are included in revised MS (Page 7-8).

Comment 5: The western blot data need quantification to further support the conclusion.

Response: The western blots of Fig 3A, D, G, J M, Fig 4I, Fig EV3B, E, H and Fig EV6D have been quantified. We thank the reviewer for bringing this to our attention.

Comment 6: Fig 1a showed that YTHDC1 significantly decreases during lung fibrosis in bleo treated mice, however, in Fig EV3b, YTHDC1 significantly increase in bleo treated cells, is this the time difference? The authors should measure the YTHDC1 level at different time points.

Response: The experiment below showed that the level of YTHDC1 is significantly increased after cells treated with BLM for short times without senescence. However, when cells were treated with BLM for long time (96h) to induce cellular senescence, the YTHDC1 protein level is significantly decreased (Picture 4). In our manuscript, the lung tissues in Fig 1a are senescent, as evidence by the Fig 1j and Fig EV1m-p. In

Fig EV3b, we treated the cells with BLM for 4h, which caused DNA damage but not cellular senescence.

Picture4: The protein level of YTHDC1 decreases after BLM-induced cell senescence

a, Immunoblot analysis of YTHDC1 in L2 cells. L2 cells were treated with BLM for indicated times prior to analysis. **b**, Quantification of panel a. The relative YTHDC1 protein level was determined by normalizing the intensity of YTHDC1 to the intensity of GAPDH. (n=3). **c**, Quantification of the SA-β-gal staining from panel a. The percentage of SA-β-gal positive cells was calculated (n ≥ 100 cells). The unpaired Student's two-tailed t-test was used to determine the statistical significance (*P<0.05, ***P<0.001, ****P<0.0001).

Comment 7: It is hard to visualize in the images provided in Fig EV1C, as the authors stated that WTAP and FTO universally express in different kinds of cells besides AECII.

Response: We apologize for confusing the reviewer by over-interpreting the data. We have removed this sentence.

References:

1. King, T.E., Jr., Pardo, A. and Selman, M. (2011) Idiopathic pulmonary fibrosis. *Lancet*, **378**, 1949-1961.
2. Lin, Y. and Xu, Z. (2020) Fibroblast Senescence in Idiopathic Pulmonary Fibrosis. *Front Cell Dev Biol*, **8**, 593283.
3. Parimon, T., Yao, C., Stripp, B.R., Noble, P.W. and Chen, P. (2020) Alveolar Epithelial Type II Cells as Drivers of Lung Fibrosis in Idiopathic Pulmonary Fibrosis. *Int J Mol Sci*, **21**.
4. Selman, M., King, T.E., Pardo, A., American Thoracic, S., European Respiratory, S. and American College of Chest, P. (2001) Idiopathic pulmonary fibrosis: prevailing and evolving hypotheses about its pathogenesis and implications for therapy. *Ann Intern Med*, **134**, 136-151.
5. Selman, M. and Pardo, A. (2020) The leading role of epithelial cells in the pathogenesis of idiopathic pulmonary fibrosis. *Cell Signal*, **66**, 109482.
6. Bartram, U. and Speer, C.P. (2004) The role of transforming growth factor beta in lung development and disease. *Chest*, **125**, 754-765.

7. Willis, B.C. and Borok, Z. (2007) TGF-beta-induced EMT: mechanisms and implications for fibrotic lung disease. *Am J Physiol Lung Cell Mol Physiol*, **293**, L525-534.
8. Arribillaga, L., Dotor, J., Basagoiti, M., Riezu-Boj, J.I., Borrás-Cuesta, F., Lasarte, J.J., Sarobe, P., Cornet, M.E. and Feijoo, E. (2011) Therapeutic effect of a peptide inhibitor of TGF-beta on pulmonary fibrosis. *Cytokine*, **53**, 327-333.
9. Giri, S.N., Hyde, D.M. and Hollinger, M.A. (1993) Effect of antibody to transforming growth factor beta on bleomycin induced accumulation of lung collagen in mice. *Thorax*, **48**, 959-966.
10. Schafer, M.J., White, T.A., Iijima, K., Haak, A.J., Ligresti, G., Atkinson, E.J., Oberg, A.L., Birch, J., Salmonowicz, H., Zhu, Y. *et al.* (2017) Cellular senescence mediates fibrotic pulmonary disease. *Nat Commun*, **8**, 14532.
11. Murray, L.A., Zhang, H., Oak, S.R., Coelho, A.L., Herath, A., Flaherty, K.R., Lee, J., Bell, M., Knight, D.A., Martinez, F.J. *et al.* (2014) Targeting interleukin-13 with tralokinumab attenuates lung fibrosis and epithelial damage in a humanized SCID idiopathic pulmonary fibrosis model. *Am J Respir Cell Mol Biol*, **50**, 985-994.
12. Aoshiba, K., Tsuji, T. and Nagai, A. (2003) Bleomycin induces cellular senescence in alveolar epithelial cells. *The European respiratory journal*, **22**, 436-443.

Dear Dr Liu,

Thank you for submitting your revised manuscript (EMBOJ-2023-113675R) to The EMBO Journal. Please accept my sincere apologies for the unusual delay in assessing your amended manuscript, which was due to protracted referee input as well as detailed discussion in the editorial team. Your amended study was sent back to the referees for their re-evaluation, and we have received comments from both of them, which I enclose below.

As you will see, referee #1 stated that the work has been substantially improved by the complementary work and s/he is now supportive of publication, while referee #2 remains overall more critical. We have concluded that in light of the strong encouragement by referee #1 that we can pursue your study further and proceed towards publication, pending minor revision addressing the remaining requests by referee #2 regarding complementary stainings to annotate specificity for the AT2 cell type.

Also, we now need you to take care of a number of minor issues related to formatting and data presentation as detailed below, which should be addressed at re-submission.

Please contact me at any time if you have additional questions related to below points.

As you might have noted on our web page, every paper at the EMBO Journal now includes a 'Synopsis', displayed on the html and freely accessible to all readers. The synopsis includes a 'model' figure as well as 2-5 one-short-sentence bullet points that summarize the article. I would appreciate if you could provide this figure and the bullet points.

Thank you for giving us the chance to consider your manuscript for The EMBO Journal. I look forward to your final revision.

Again, please contact me at any time if you need any help or have further questions.

Kind regards,

Daniel Klimmeck

>> This should be the order of the sections in the manuscript: abstract, introduction, results, discussion, materials & methods, data availability section, acknowledgments, disclosure statement and competing interests, references, main figure legends, expanded figure legends

>> Adjust the title of the 'Declaration of Interests' section to 'Disclosure and Competing Interests Statement'.

>> Adjust the title of the current 'Quantification and statistical analysis' section to 'Statistical analysis'.

>> Author Contributions: Please remove the author contributions information from the manuscript text. Note that CRediT has replaced the traditional author contributions section as of now because it offers a systematic machine-readable author contributions format that allows for more effective research assessment. and use the free text boxes beneath each contributing author's name to add specific details on the author's contribution.

More information is available in our guide to authors.

>> Funding information: please update the information provided in our online system. Currently missing: the Guangdong Basic and Applied Basic Research Foundation (2021A1515110989).

>> Reference format: please update, 'et al' should be used after 10 author names.

>>Data citation: amend the dataset reference Ahangari et al 2019 with '[DATASET]'

>> Callouts: add and adjust order of callouts for figure 1F in the main text.

>> EV Figures: please limit the number of EV figures to maximally five. Upload as Figures.

>> As to our journal policies we kindly request uncropped source data for figures 3A and EV4 C

>> Consider additional changes and comments from our production team as indicated by the .doc file enclosed and leave changes in track mode.

We realize that it is difficult to revise to a specific deadline. In the interest of protecting the conceptual advance provided by the work, we recommend a revision within 3 months (24th Oct 2023). Please discuss the revision progress ahead of this time with the editor if you require more time to complete the revisions.

Referee #1:

Authors propose that YTHDC1 counteracts pulmonary senescence and disease. They show that YTHDC1 primarily expresses in pulmonary alveolar epithelial type 2 (AECII) cells and when mice are treated with bleomycin (a model of IPF), YTHDC1 expression is significantly decreased in AECII. Authors then suggest that YTHDC1 depletion accelerates senescence and fibrosis in the lung after BLM and its overexpression alleviates pulmonary senescence and fibrosis. Finally, authors conduct mechanistic studies which propose that YTHDC1 promotes the interaction between TopBP1 and MRE11, thus activating the ATR and facilitating repair of DNA damage. Results are interesting and the mechanism described novel.

Authors have made a considerable effort to answer my concerns and I feel that the manuscript has improved substantially.

Quality of the data is generally good. I believe the conclusions are supported by the data. I recommend publication.

A minor weakness was lack of IPF patient data and the limitations inherent to the bleomycin model, however, I believe despite this- the mechanisms described are interesting and the science is sound.

Referee #2:

The authors have not addressed my points well.

For comment 1, to further confirm the cell type in which YTHDC1 was expressed, primary AT2 cells and other cells like lung fibroblasts from mice treated with BLM can be isolated and measure the expression.

For comment 2, multiple reports have showed that 14 days after bleomycin instillation is the best time point to measure lung fibrosis parameters such as Masson's staining, as day 7 is still in the inflammatory phase and very mild damage can be seen.

For comment 4, P21 and P16 should be stained with AT2 cell marker and YTHDC1 to better characterize the role of YTHDC1 in AT2 cell senescence.

Comment 6, the authors have not addressed the reason for the difference.

Comment 7, the IF staining quality is poor to get any conclusion, also for the IF images for most cell cultures, only one cells in showing in the field.

Referee #1:

Authors propose that YTHDC1 counteracts pulmonary senescence and disease. They show that YTHDC1 primarily expresses in pulmonary alveolar epithelial type 2 (AECII) cells and when mice are treated with bleomycin (a model of IPF), YTHDC1 expression is significantly decreased in AECII. Authors then suggest that YTHDC1 depletion accelerates senescence and fibrosis in the lung after BLM and its overexpression alleviates pulmonary senescence and fibrosis. Finally, authors conduct mechanistic studies which propose that YTHDC1 promotes the interaction between TopBP1 and MRE11, thus activating the ATR and facilitating repair of DNA damage. Results are interesting and the mechanism described novel.

Authors have made a considerable effort to answer my concerns and I feel that the manuscript has improved substantially. Quality of the data is generally good. I believe the conclusions are supported by the data. I recommend publication.

A minor weakness was lack of IPF patient data and the limitations inherent to the bleomycin model, however, I believe despite this- the mechanisms described are interesting and the science is sound.

We thank the reviewer for positive comment on revision. Regarding the lack of IPF patient data and the inherent limitations of the bleomycin model, we will be more diligent in addressing these aspects in our future studies. We would like to express our gratitude once more to the reviewers for their valuable suggestions.

Referee #2:

The authors have not addressed my points well.

For comment 1, to further confirm the cell type in which YTHDC1 was expressed, primary AT2 cells and other cells like lung fibroblasts from mice treated with BLM can be isolated and measure the expression.

Response: As suggested, we isolated the AECII cells (the purity is more than 90%) and fibroblasts (the purity is more than 95%) from mice lung treated with or without BLM and measure the expression level of YTHDC1. We found that the level of YTHDC1 decreases in AECII cells but not the fibroblasts cells (Appendix Figure S1), which is consistent with the conclusion of Figure 1A and suggests that YTHDC1 plays an important role in AECII cells during pulmonary senescence.

For comment 2, multiple reports have showed that 14 days after bleomycin instillation is the best time point to measure lung fibrosis parameters such as Masson's staining, as day 7 is still in the inflammatory phase and very mild damage can be seen.

Response: We thanks the reviewer for the suggestion. Masson's staining and α -SMA IHC have been performed to measure lung fibrosis in mice lung transfected with shYTHDC1 AAV or YTHDC1 overexpressing AAV 14 days after BLM treatment (Appendix Figure S2). These results are consistent with and strengthen the conclusion obtained from mice lung treated with BLM for 7 days. We thank the reviewer for his/her suggestion.

For comment 4, P21 and P16 should be stained with AT2 cell marker and YTHDC1 to better characterize the role of YTHDC1 in AT2 cell senescence.

Response: We thanks the reviewer for this suggestion. IF experiments have been performed to stain the p21 and p16 with SPC (the marker of AECII cells) and YTHDC1 in mice lung transfected with YTHDC1 overexpressing AAV (Figure EV2 H-M). As shown in the results, overexpression of YTHDC1-WT and -MUT specifically decreased the level of p21 and p16 in AEC2 cells but not in other cells,

which support the idea that YTHDC1 plays a protective role in ATII cells during pulmonary senescence.

Comment 6, the authors have not addressed the reason for the difference.

Original Comment 6: Fig 1a showed that YTHDC1 significantly decreases during lung fibrosis in bleo treated mice, however, in Fig EV3b, YTHDC1 significantly increase in bleo treated cells, is this the time difference? The authors should measure the YTHDC1 level at different time points.

Original Response: The below experiment showed that the level of YTHDC1 is significantly increased after treated the cells with BLM for short times without senescence. However, when we treated the cells with long time (96h) to induce cellular senescence, the YTHDC1 protein level is significantly decreased compared with 0h group. In our manuscript, the lung tissue in Figure 1a is senescence, as evidence by the Figure 1 j and Figure EV1m-p (The NC group without or with BLM). In Figure EV3b, we treated the cells with BLM for 4h, which caused DNA damage

but no the cellular senescence.

Picture4: The protein level of YTHDC1 decreases after BLM-induced cell senescence

a, Immunoblot analysis of YTHDC1 in L2 cells. L2 cells were treated with BLM for indicated times prior to analysis. b, Quantification of panel a. The relative YTHDC1 protein level was determined by normalizing the intensity of YTHDC1 to the intensity of GAPDH. (n=3). c, Quantification of the SA-β-gal staining from panel a. The percentage of SA-β-gal positive cells was calculated (n ≥ 100 cells). The unpaired Student's two-tailed t-test was used to determine the statistical significance (*P<0.05, ***P<0.001, ****P<0.0001).

Response: As shown in the original response and the results in the MS, we found that short-term DNA damage increased YTHDC1 expression, whereas long-term sustained DNA damage decreased YTHDC1 expression. This phenomenon is common in DNA damage response, such as p53. The expression of p53 is activated upon DNA damage,

triggering various cellular responses, including cell cycle arrest and DNA repair. However, prolonged treatment with DNA damage drugs lead to reduction of p53 through ubiquitination and proteasomal degradation. Therefore, the level of p53 initially increases and then subsequently decreases during prolonged DNA damage treatment (Di Leonardo, Linke et al., 1994, Hirao, Kong et al., 2000). There are other proteins that also exhibit similar expression pattern, such as ATM(Lavin, 2007), and YTHDC1 may be one of them. The potential regulatory mechanism may be related to phosphorylation or ubiquitination of YTHDC1. While these aspects were not investigated in this study, they pose intriguing questions to explore in the future research.

Comment 7, the IF staining quality is poor to get any conclusion, also for the IF images for most cell cultures, only one cells in showing in the field.

Response: As suggested, the IF in Figure EV1C has been revised for better visualization of protein localization. The IF in Figure 4E, Figure EV4A, Figure EV5 A, D and H (Figure EV 6B in original MS) also have been revised to show multiple cells in the field.

Di Leonardo A, Linke SP, Clarkin K, Wahl GM (1994) DNA damage triggers a prolonged p53-dependent G1 arrest and long-term induction of Cip1 in normal human fibroblasts. *Genes Dev* 8: 2540-51

Hirao A, Kong YY, Matsuoka S, Wakeham A, Ruland J, Yoshida H, Liu D, Elledge SJ, Mak TW (2000) DNA damage-induced activation of p53 by the checkpoint kinase Chk2. *Science* 287: 1824-7

Lavin MF (2007) ATM and the Mre11 complex combine to recognize and signal DNA double-strand breaks. *Oncogene* 26: 7749-58

Dear Dr Liu,

Thank you for submitting the revised version of your manuscript. I have now evaluated your amended manuscript and concluded that the remaining minor concerns have been sufficiently addressed.

Thus, I am pleased to inform you that your manuscript has been accepted for publication in the EMBO Journal.

Please note that it is EMBO Journal policy for the transcript of the editorial process (containing referee reports and your response letter) to be published as an online supplement to each paper. I would thus like to ask for your consent on keeping the additional referee figures included in this file.

Also, in case you might NOT want the transparent process file published at all, you will also need to inform us via email immediately. More information is available here:

<https://www.embopress.org/page/journal/14602075/authorguide#transparentprocess>

Please note that in order to be able to start the production process, our publisher will need and contact you shortly regarding the page charge authorisation and licence to publish forms.

Authors of accepted peer-reviewed original research articles may choose to pay a fee in order for their published article to be made freely accessible to all online immediately upon publication. The EMBO Open fee is fixed at \$6,540 USD / £5,310 GBP / €5,900 EUR (+ VAT where applicable), pending application of a waiver which might be applicable in this case as discussed.

Should you be planning a Press Release on your article, please get in contact with embojournal@wiley.com as early as possible, in order to coordinate publication and release dates.

On a different note, I would like to alert you that EMBO Press is currently developing a new format for a video-synopsis of work published with us, which essentially is a short, author-generated film explaining the core findings in hand drawings, and, as we believe, can be very useful to increase visibility of the work. This has proven to offer a nice opportunity for exposure i.p. for the first author(s) of the study. Please see the following link for representative examples and their integration into the article web page:

<https://www.embopress.org/doi/full/10.15252/emboj.2019103932>

If you have any questions, please do not hesitate to call or email the Editorial Office.

Kind regards,

Daniel Klimmeck

Daniel Klimmeck, PhD
Senior Editor
The EMBO Journal
EMBO
Postfach 1022-40
Meyerhofstrasse 1
D-69117 Heidelberg
contact@embojournal.org
Submit at: <http://emboj.msubmit.net>